# A scalable *Drosophila* assay for clinical interpretation of human *PTEN* variants in suppression of PI3K/AKT induced cellular proliferation

**Payel Ganguly**[1‡], **Landiso Madonsela**[2‡], **Jesse T. Chao**[1], **Christopher J. R. Loewen**[1], **Timothy P. O'Connor**[1], **Esther M. Verheyen**[2]*, **Douglas W. Allan**[1]*

1 Department of Cellular and Physiological Sciences, Life Sciences Institute, Djavad Mowafaghian Centre for Brain Health, University of British Columbia, Vancouver, British Columbia, Canada, 2 Department of Molecular Biology and Biochemistry, Centre for Cell Biology, Development and Disease, Simon Fraser University, Burnaby, British Columbia, Canada

‡ These authors share first authorship on this work.
* everheye@sfu.ca (EMV); doug.allan@ubc.ca (DWA)

**Data Availability Statement:** Relevant data are within the paper and its Supporting information files. Visit https://github.com/jessecanada/PTEN_

## Abstract

Gene variant discovery is becoming routine, but it remains difficult to usefully interpret the functional consequence or disease relevance of most variants. To fill this interpretation gap, experimental assays of variant function are becoming common place. Yet, it remains challenging to make these assays reproducible, scalable to high numbers of variants, and capable of assessing defined gene-disease mechanism for clinical interpretation aligned to the ClinGen Sequence Variant Interpretation (SVI) Working Group guidelines for 'well-established assays'. *Drosophila melanogaster* offers great potential as an assay platform, but was untested for high numbers of human variants adherent to these guidelines. Here, we wished to test the utility of *Drosophila* as a platform for scalable well-established assays. We took a genetic interaction approach to test the function of ~100 human PTEN variants in cancer-relevant suppression of PI3K/AKT signaling in cellular growth and proliferation. We validated the assay using biochemically characterized PTEN mutants as well as 23 total known pathogenic and benign PTEN variants, all of which the assay correctly assigned into predicted functional categories. Additionally, function calls for these variants correlated very well with our recent published data from a human cell line. Finally, using these pathogenic and benign variants to calibrate the assay, we could set readout thresholds for clinical interpretation of the pathogenicity of 70 other PTEN variants. Overall, we demonstrate that *Drosophila* offers a powerful assay platform for clinical variant interpretation, that can be used in conjunction with other well-established assays, to increase confidence in the accurate assessment of variant function and pathogenicity.

Fly to view all datasets used in this study and our Jupyter notebooks for codes, instructions, and detailed analytics.

**Funding:** Work in the Allan and O'Connor labs was supported by a Simons Foundation for Autism Research Initiative, Award #573845 (2018-2021): "A multi-model screening approach for the functional characterization of large numbers of ASD variants" to D.A., T.O. and C.L. Work in the Verheyen lab was supported by a research grant from the Natural Sciences and Engineering Council of Canada (NSERC), grant number RGPIN/2014-05749 to E.V. Work in the Loewen lab was supported by Canadian Institutes for Health Research award # PJT-152967 to C.L. The funders had no role in study design, data collection and analysis, decision to publish, or preparation of the manuscript.

**Competing interests:** The authors have declared that no competing interests exist.

## Author summary

DNA sequencing is becoming commonplace in the clinic, as physicians read your DNA to determine if you have variations in gene sequence that may help with diagnosis and therapy planning. The assumption is that we would know a damaging sequence variation if we saw it. However, this is in fact extremely difficult, and to this day most sequence variations observed in people have unknown disease implication. We must turn to experimental assessments of the variant's impact on gene function. This current report shows that an organism widely used to understand human disease mechanisms, *Drosophila melanogaster*, is a valid option for testing the function of hundreds of variants in a human gene of interest. In this case, we tested the function of a hundred variants in the human *PTEN* gene, found in patients with cancer or autism spectrum disorder, and we were able to pinpoint which ones are likely to contribute to disease, and which were not. Our work provides evidence that *Drosophila* offers a powerful experimental platform for establishing assay to easily test the function of high numbers of gene variations, that can be used to complement and extend other similar assays.

## Introduction

Discovery of disease-causing genetic variants has long served as a powerful tool in understanding and tackling disease. Exome and genome sequencing is becoming increasingly routine in clinical practice, and is finding variants at an increasing pace [1, 2]. The underlying assumption is that the discovery of pathogenic variants will provide insight into disease aetiology and can guide precision therapy [3–5]. However, in spite of the many computational tools for variant effect prediction [6–11], making confident predictions of pathogenicity remains a challenge for many variants, and most remain 'variants of uncertain significance' (VUS) [12, 13]. To fill this interpretation gap, a wide variety of functional assays are being developed [14, 15].

Recently, the Clinical Genome Resource (ClinGen) Sequence Variant Interpretation (SVI) Working Group provided guidelines for setting up 'well-established' assays for clinical variant interpretation [12]. A critical consideration is how to translate a variant's relative function into a clinical interpretation in an assay, reliably and reproducibly. Key criteria include: (i) Assessing variant function in an assay modeling a defined gene-disease mechanism. (ii) Calibrating the assay with a minimum of 11 known pathogenic and benign variants, preferably established on clinical or approved grounds. (iii) Assays must be optimized for rigor and reproducibility.

*Drosophila* has tremendous potential as a complementary platform for clinical variant interpretation at the scale of hundreds of variants. Yet, to our knowledge, no more than 12 human variants for one gene have ever been tested in a *Drosophila* study [16, 17], and none have been calibrated against enough pathogenic and benign variants. Key challenges include testing defined gene-disease mechanisms, and making the assay scalable which preferentially requires an easily scored phenotype in the F1 progeny of a single cross. *Drosophila* offers tremendous flexibility in establishing these conditions. While different at a gross anatomical level, most human organ systems and molecular pathways are well conserved [18, 19]. Moreover, *Drosophila* offers an experimentally tractable platform, including a versatile and cost-effective suite of molecular genetic tools, and a wealth of annotated genetic and protein interactions for thousands of genes [20–22]. These establish a solid foundation for developing genetic interaction strategies for explicit testing of defined gene-disease mechanisms. Additionally, integrase-

based transgenesis assures that all human variants can be efficiently generated for expression at the same level in specific tissues of interest, or in an orthologous gene replacement strategy [23–26].

The goal of our study was to assess the efficacy of *Drosophila* in well-established assays for scalable variant functionalization and clinical variant interpretation of human coding variants. We use PTEN as an example. *PTEN* is a highly penetrant autosomal dominant cancer predisposition gene; haploinsufficiency or partial loss of *PTEN* tumor suppressor activity is observed in sporadic and heritable cancers [27–30]. PTEN hamartoma syndrome (PHTS), which can arise in childhood, and encompasses a variety of germline disorders such as Cowden and Bannayan-Riley-Ruvalcaba (BRR) syndromes [31–33], as well as macrocephaly, epilepsy, mental retardation/developmental delay and autism spectrum disorder (ASD) [34–36]. PTEN is a dual lipid and protein phosphatase that acts as a crucial repressor of the Phosphoinositide 3-kinase (PI3K) /Protein kinase B (PKB/AKT) pathway [37–39]. Increased PI3K/AKT pathway activity is a major contributor to human cancer [40]. PI3K catalyzes the conversion of phosphatidylinositol 4,5-bisphosphate ($PIP_2$) to phosphatidylinositol-3, 4, 5-triphosphate ($PIP_3$), leading to AKT phosphorylation by phosphoinositide-dependent kinase-1 (PDK1) [41–44]. This promotes cellular survival, proliferation and growth. PTEN's lipid phosphatase activity converts $PIP_3$ to $PIP_2$, thus suppressing PI3K/AKT signaling.

Identifying PTEN variants with reduced function is considered clinically actionable (ClinGen), and is an important contributory criterion for a PHTS diagnosis, typically resulting in a recommendation of surveillance and screening of related individuals [36, 45, 46]. In both PHTS and ASD, early detection and intervention are viewed as the most effective management strategies for both ASD or *PTEN*-associated cancers [34, 45–49]. However, as of last database access (November 22, 2020), ClinVar classified 412 of the annotated 575 missense PTEN variants as VUS, and the COSMIC catalog of somatic mutations records 2492 missense PTEN variants and the gnomAD records 84 missense PTEN variants for which there is no easy way to assess their function. Thus, there is a pressing need to create robust, inexpensive, efficient assays for PTEN variant functionalization.

The assay we developed exploits a genetic interaction approach which tests the established role of human PTEN as a suppressor of PI3K-induced increases in PIP3 and pAKT levels, as well as cellular proliferation and tissue growth. In this context, we tested the relative function of ~100 human PTEN variants. We successfully calibrated the assay against 23 total known pathogenic and benign variants, and 4 biochemically characterized variants. This shows the assay provides high positive predictive value for pathogenic variants and high negative predictive value for benign variants. Also, the assay performed very well when benchmarked against variant function data acquired from our previous report using a human cell line [50]. Overall, we validate the utility of *Drosophila* in providing reliable and reproducible functional data for high numbers of human variants, in assays conforming to guidelines for clinical interpretation. Notably, as an *in vivo* genetic model with efficient integrase-based transgenesis for locus-specific integration of human variants [23–25], the model offers tremendous reproducibility. Additionally, the ease of establishing a wide variety of sensitized genetic backgrounds in *Drosophila* tissues of interest provides a solid foundation for developing assays tuned to defined gene-disease mechanisms. Therefore, as confident assessments of variant function are best achieved through consensus across numerous well-established assays [14, 50], we propose that *Drosophila* assays offer a powerful and flexible option for complementing and extending other well-established assays for clinical variant interpretation.

## Materials and methods

### *Drosophila* strains

Flies were maintained on standard cornmeal food at room temperature. Crosses were raised at 70% humidity and 25˚C, unless otherwise noted. The following lines were obtained from the Bloomington *Drosophila* Stock Center. *P{w[+mC] = Dp110-CAAX}1, y[1] w[*]* (BL25908) (referred to as *PI3K^{act}*). *P{UAS-GFP.U}1, y[1] w\* P{GawB}bi^{omb-Gal4} [omb-GAL4]* (BL58815) (referred to as *omb-GAL4, UAS-GFP,* or simply *omb-GAL4*). *y[1] w[*]; P{w[+mW.hs] = en2.4-Gal4}e22c; P{w[+mC] = tGPH}4/TM3, Ser[1]* (BL8165). *Pten^{117}* was a kind gift from Hugo Stocker, ETH Zurich, Switzerland [51]. *Pten^{100}* was a kind gift from Elizabeth Rideout, UBC, Vancouver, Canada [52]. All *UAS-PTEN* stocks were previously integrated into the *attP2* locus of the *Drosophila* genome [23, 50, 53]. Integration into the same site ensures reproducible expression of all variants to allow for direct comparison of *PTEN* variant function in *Drosophila* tissues. For all crosses, the parental *attP2* containing stock was used as a control. Constitutively active *Drosophila* PI3K (*PI3K92E-CAAX,* referred to as *PI3K^{act}*) [54], was recombined onto the same chromosome as *omb-GAL4* and *UAS-GFP*. The recombinant line is referred to as *omb-GAL4>UAS-PI3K^{act}/FM7,* or *omb>PI3K^{act}*.

### Pupal volume measurement

Fly crosses were maintained on standard cornmeal food at room temperature. For every cross, the same number of parental flies were crossed and maintained for 36 hours prior to being transferred to grape juice/agar plates, for serial 24-hour egg collection windows. To measure pupal volume, larvae from different genotypes were synchronized such that they were collected in the early L1 stage and cultured under the same controlled conditions (50 larvae/vial) to avoid crowding [55]. Larvae were genotyped against a GFP-expressing balancer. The length and diameter of *Pten* heterozygote controls (*Pten^{100}/+;da-GAL4/attP2*), *Pten* heteroallelic mutants (*Pten^{100}/Pten^{117};da-GAL4/attP2*) and *Pten* rescue (*Pten^{100}/Pten^{117};da-GAL4/UAS-PTEN-WT*) was measured using ImageJ and the pupal volume was calculated by using the formula $4/3\pi(L/2)\,(l/2)^2$ (L, length; l, diameter) [55].

### Immunohistology

Larval wing imaginal discs were fixed in 4% Paraformaldehyde (15 or 30 min, RT), followed by 2 rinses in 1X Phosphate buffer saline with 0.1% Tween 20 (0.1% PBT) and then washed in 0.1% PBT three times (5 min, 10 min, 15 min, RT), blocked in 2% bovine serum albumin or 5% donkey serum in 0.1% PBT (1 hr, RT), incubated with primary antibodies overnight (4˚C), washed three times in 0.1% PBT, and incubated with secondary antibodies (1 hr 40 min, RT) and washed 3 times. Primary antibodies used: chicken α-GFP (1:1000; ab13970, Abcam, Ontario, Canada), rabbit α-*Drosophila* phospho-Ser505 Akt (1:200; #4054, Cell Signaling Technology, Danvers, MA), mouse α-human PTEN (1:50; #A487, R&D systems, Minneapolis, MN), rabbit α-*Drosophila* phospho Histone 3 (1:1000; #9701s, Cell Signaling Technology #9701S, Danvers, MA). Secondary antibodies used were: donkey anti-chicken 488 (1:700), donkey anti-rabbit Cy3 (1:500), donkey anti-mouse Cy5 (1:500) and donkey anti-rabbit 647 (1:500, Jackson ImmunoResearch, West Grove, PA).

### Adult wing imaging and data analysis for PTEN variants

The relative function of PTEN variants [indicated by the amino acid number and substitution] was assessed using wing size as an assay across seven independent experimental batches (S1 Table). Each batch of crosses comprised flies in which a different random subset of variants

was tested; *PTEN-WT* and *attP2* were also included in every group of crosses, to control for any batch effects. Stocks were assigned numbers that did not reveal variant identity to assure blinding in scoring crosses. For every cross, the same number of males and females were utilized, under the same culture conditions. A single wing was dissected from each adult progeny at day 13 or 14, then stored in 70% ethanol until slide-mounting in Aquatex mounting solution (EMD Chemicals, USA). A minimum of 10 wings were mounted per genotype. Adult wings were imaged with an Axioplan-2 microscope (Zeiss, Oberkochen, Germany) at 5X and 20X magnification. Wing size areas were calculated using Adobe Photoshop CS3 (Adobe Systems, San Jose, CA).

Data processing: All data points (i.e. each single dot) were normalized to the mean wing size of the control, $omb>PI3K^{act}+attP2$ (= 1). Then each dataset was normalized using the following formula:

$$\text{Normalize wing area} = \frac{omb > PI3K^{act} + hPTEN \ variant \ wing \ area}{Mean \ omb > PI3K^{act} \ + \ attP2 \ wing \ area}$$

Wing hair counts (number) were calculated using ImageJ. Wing hairs were counted in a square of fixed size just above the posterior cross vein in the interval between longitudinal veins 3 and 4. The hairs were counted in each genotype in the same position and between 9–11 wings were measured, per genotype.

## Quantification of GFP area and PH3 staining

Images of L3 wing discs were taken on a Nikon Air laser-scanning confocal microscope (Nikon, Tokyo, Japan). To quantify GFP area and PH3 staining, all tissues were processed with the same reagents, imaged, and analyzed in identical ways. The GFP area was calculated using ImageJ and reported as a ratio of the whole disc area to account for differences in wing disc size. PH3 cell counting was performed on max-projection images using ImageJ software. A minimum of 10 discs were quantified per genotype. Statistical analyses and graphing of datasets were performed using GraphPad Prism 8 (GraphPad Software, San Diego, CA) and data within graphs were compared using one-way ANOVA followed by Tukey *post hoc* analysis. Differences between groups were considered statistically significant when $p<0.05$. Data are presented as mean ± Standard Deviation (SD).

## Quantification of pAkt levels

Quantification of pAkt immunoreactivity was performed on each L3 wing disc. In all cases, six or more wing discs were dissected and imaged for each genotype. Images were acquired with a Zeiss LSM 880 with a 34-channel spectral Quasar detector. Representative images of wing discs were processed in Adobe Photoshop CS5 (Adobe Systems, San Jose, CA). To quantify pAkt levels, all tissues were processed with the same reagents, and then imaged and analyzed in identical ways. For each image, 3 z-stacks were selected immediately under the peripodial cell layer from the apical region of each wing disc, to create a maximum projection image that was quantified using Image J. The pAkt intensity levels were reported as a posterior/anterior intensity ratio, where an identical square was drawn within each posterior and anterior compartment of the wing disc to measure the intensity in the same region. Statistical analysis and graphs of datasets were performed using GraphPad Prism 8 (GraphPad Software, San Diego, CA) and data within graphs were compared using one-way ANOVA followed by Tukey *post hoc* analysis. Differences between groups were considered statistically significant when $p<0.05$, since ANOVA corrects for multiple testing. Data are presented as mean ± Standard Deviation (SD).

## Computational analysis

**Computation requirements.** Bioinformatic analyses were done using Python 3.6 and the following packages: numpy 1.18.1, pandas 1.0.3 and matplotlib 3.1.3. Visit https://github.com/jessecanada/PTEN_Fly to view all datasets used in this study and our Jupyter notebooks for codes, instructions, and detailed analytics.

**Data processing and normalization.** Datasets were acquired as follows: PolyPhen-2 and CADD prediction results were parsed from the Ensembl Genome Browser [56, 57]. HEK [50] and yeast [58] data were obtained from their publications. ClinVar classifications, gnomAD and COSMIC data were downloaded from their respective database [59–61]. The last date of access was November 22, 2020.

For the fly wing dataset, we normalized each variant (norm) such that the *no PTEN (attP2)* control = 0 and the *PTEN-WT* control = 1 within its respective batch to account for experimental batch effects.

$$norm = \frac{Var - attP2}{WT - attP2}$$

Additionally, we propagated the uncertainty by summing the errors in quadrature,

$$relative\ uncertainty\ of\ normalized\ wing\ data\ (\frac{\delta norm}{|norm|}) =$$

$$\sqrt{(\frac{\delta Var}{var} + \frac{\delta attP2}{attP2})^2 + (\frac{\delta WT}{WT} + \frac{\delta attP2}{attP2})^2}$$

**Data visualization.** Hierarchical clustering was performed using complete linkage. To visualize data as heatmaps, we capped variant functional scores above 1.0 (*WT*) at 1.2, and scores below 0.0 (*no PTEN*) at -0.2.

## Results

### Human *PTEN* rescues a *Drosophila Pten* hypomorph overgrowth phenotype

As a first step towards characterizing *PTEN* variants, we sought to establish that the annotated wild type *PTEN* (*PTEN-WT*) could functionally replace *Pten* in *Drosophila* tissues, when integrated into the *attP2* locus. Two previous studies show differing phenotypic results for *PTEN* overexpression in the eye disc, but neither tested if *PTEN* could rescue loss of *Drosophila Pten* [62, 63]. *Pten* nulls are lethal [64], therefore we chose to use a strong hypomorphic heteroallelic combination, $Pten^{100}/Pten^{117}$, that exhibits increased larval growth leading to increased pupal volume and adult weight [52] (Fig 1A and 1D). Thus, we could be sure we were testing a growth phenotype, rather than lethality which is reported to have contributions from Pten regulation of the actin cytoskeleton as well as cellular growth mechanisms [64]. We examined the capacity for *PTEN* to rescue the overgrowth phenotype, examining males and females separately due to their difference in size. *UAS-PTEN-WT* was expressed using the ubiquitous *daughterless (da)-GAL4* driver in developing embryos and larvae. We confirmed that $Pten^{100}/Pten^{117}$; *da-GAL4/attP2* mutants had significantly greater pupal volume and adult fly weight than did $Pten^{100}/+$; *da-GAL4/attP2* heterozygous controls (Fig 1A and 1D). In $Pten^{100}/Pten^{117}$; *da-GAL4/UAS-PTEN-WT* animals, we observed a rescue of mutant pupal volume and adult fly weight to control levels, in both sexes (Fig 1A–1F).

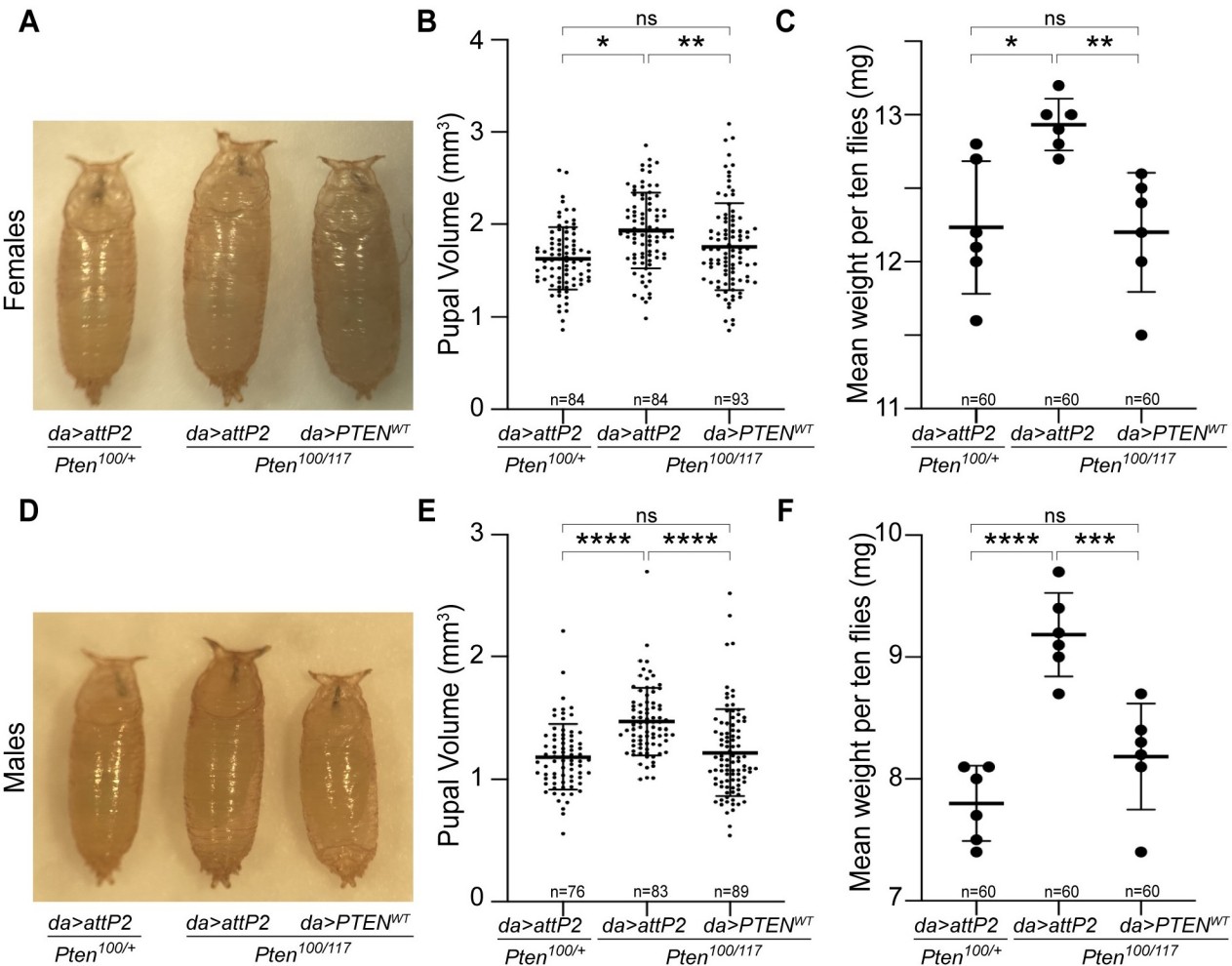

**Fig 1. Ubiquitous expression of human PTEN rescues overgrowth of *Drosophila Pten* hypomorphic mutants.** Representative images of female pupae (**A**) and male pupae (**D**) of *Pten* heterozygotes (*Pten100/+; da-GAL4/attp2*), *Pten* mutants (*Pten100/Pten117*; *da-GAL4/attp2*), and *Pten* mutants rescued by *PTEN* (*Pten100/Pten117*; *da-GAL4/UAS-PTEN-WT*). Graphs showing female and male pupal volume (**B, E**), as well as female and male body weight (**C, F**) for each genotype shown. Sample size (n) is indicated for each genotype. Each datum point within scatter plots represents a single pupa (**B, E**) or the mean of 10 adult flies (**C, D**). Expression of *PTEN-WT* in *Pten* strong hypomorphs rescued pupal volume and adult body weight in both sexes. Data expressed as mean ± SD and analyzed using one-way ANOVA with Tukey HSD *post-hoc*; * p < 0.05, ** p < 0.01, *** p < 0.001, **** p < 0.0001, ns = not significant.

Thus, *PTEN-WT* integrated into the *attP2* locus rescued the overgrowth phenotype found in *Pten* hypomorphic mutants.

## Human PTEN suppresses PI3K induced phenotypes in adult *Drosophila* wings

We next wished to establish an efficient, scalable assay for high volume assessment of PTEN variants in PI3K/AKT signaling promoting tissue growth. Building *UAS-PTEN* variants into a *Pten* hypomorphic or null background would limit the scalability of the assay, and would not explicitly test the defined gene-disease mechanism. Therefore, we decided to develop a genetic interaction approach in which PTEN variants would be assessed for their ability to suppress PI3K signaling. A genetic suppression approach is preferred in genetic interaction tests because typically only genes acting within the upregulated pathway are expected to selectively

suppress the resulting phenotype. Expression of wildtype or activated *Drosophila* Phosphatidy-linositol 3-kinase 92E (PI3K92E) increases *Drosophila* wing size by increasing both cell size and cell proliferation [54, 65]. This increased wing size provides continuous quantitative data to report the ability of PTEN to suppress PI3K92E induced wing overgrowth, as was previously shown when *Drosophila Pten* was co-expressed with wildtype PI3K92E [64]. Activated membrane targeted *PI3K92E-CAAX* (*PI3K^act*) was expressed, together with *UAS-GFP*, in the wing imaginal disc using the *omb-GAL4* driver (a wing disc pouch-specific *GAL4* driver) and adult wing area was measured (Fig 2A). Expression of PI3K^act resulted in an enlarged wing size compared to the control flies *omb-GAL4, UAS-GFP;;attP2* (Fig 2A and 2B). We tested if PTEN-WT could suppress tissue overgrowth caused by PI3K^act. We crossed *UAS-PTEN-WT* to a recombinant strain carrying *omb-GAL4, UAS-GFP, UAS-PI3K^act* (*omb>PI3K^act*). Importantly, PTEN-WT fully suppressed the *omb>PI3K^act* induced increase in adult wing size (Fig 2A and 2B). Wings of *omb>PI3K^act+PTEN-WT* animals were significantly but modestly smaller than control *omb>attP2* wings, indicating that PTEN-WT fully suppressed the hyperactivated PI3K/AKT activity and to some extent endogenous signaling. This is corroborated by our finding that PTEN-WT reduces wing size in the absence of *PI3K^act*, indicating an ability to suppress endogenous signaling (S1A and S1B Fig). These data show that PTEN suppresses PI3K-induced wing overgrowth to a more wildtype state, indicating that PTEN overexpression is not toxic.

We then set out to test four biochemically characterized PTEN variants. PTEN is a dual-specificity lipid and protein phosphatase; these established 'biochemical' variants selectively disrupt these lipid and/or protein phosphatase activities. C124S is a 'likely pathogenic' variant described in ClinVar and COSMIC, which lacks both protein and lipid phosphatase activity [38] and has apparent antimorphic activity in mouse models [66]. G129E is a pathogenic variant described in ClinVar and COSMIC, which selectively lacks lipid phosphatase activity [67] and has apparent antimorphic activity in mouse models [66]. Y138L is an engineered variant that is reported to selectively lack protein phosphatase activity [68], and 4A is an engineered variant with a 4x alanine repeat within C-terminal sequences that overrides protein autoinhibitory activity to produce a hypermorphic PTEN variant [69].

Both the C124S dual phosphatase dead and the G129E lipid phosphatase dead variants failed to suppress PI3K^act-increased wing size; they were not significantly different in size from the *attP2* control crossed to *omb>PI3K^act* (Fig 2C and 2D). This corresponds with their inability to suppress PI3K activity. The 4A hypermorphic variant severely reduced wing size significantly below control size (Fig 2C and 2D). This is consistent with its hypermorphic activity suppressing transgenically-enhanced PI3K/AKT pathway, and likely also endogenous PI3K/AKT pathway activity. The Y138L variant partially suppressed the PI3K^act-induced enhancement of wing size and was significantly different from both *attP2* controls and PTEN-WT (Fig 2C and 2D). Thus, PTEN-dependent suppression of PI3K/AKT tissue growth in *Drosophila* is absolutely dependent upon lipid phosphatase activity, and also significantly dependent on protein phosphatase activity. This indicates that this assay may be used in screening variants in which loss of these two critical functions could result in reduced PTEN function.

Finally, we compared the relative bioactivity of Pten and PTEN in *Drosophila* tissues (S1 Fig). We overexpressed PTEN-WT and Pten using *omb-GAL4* and quantified wing size in backgrounds that did or did not co-express PI3K^act (S1A–S1D Fig). In both contexts, Pten induced a much greater reduction in wing size than PTEN-WT. We also tested C124, G129E and G127R in the context of a wing with no induced PI3K^act, and found that all three failed to reduce wing size to the extent of PTEN-WT, and only in the case of G129E did we observe a weak but significant reduction in wing size compared to the *attP2* control (S1B Fig). These data indicate that, in the developing *Drosophila* wing, PTEN can suppress endogenous

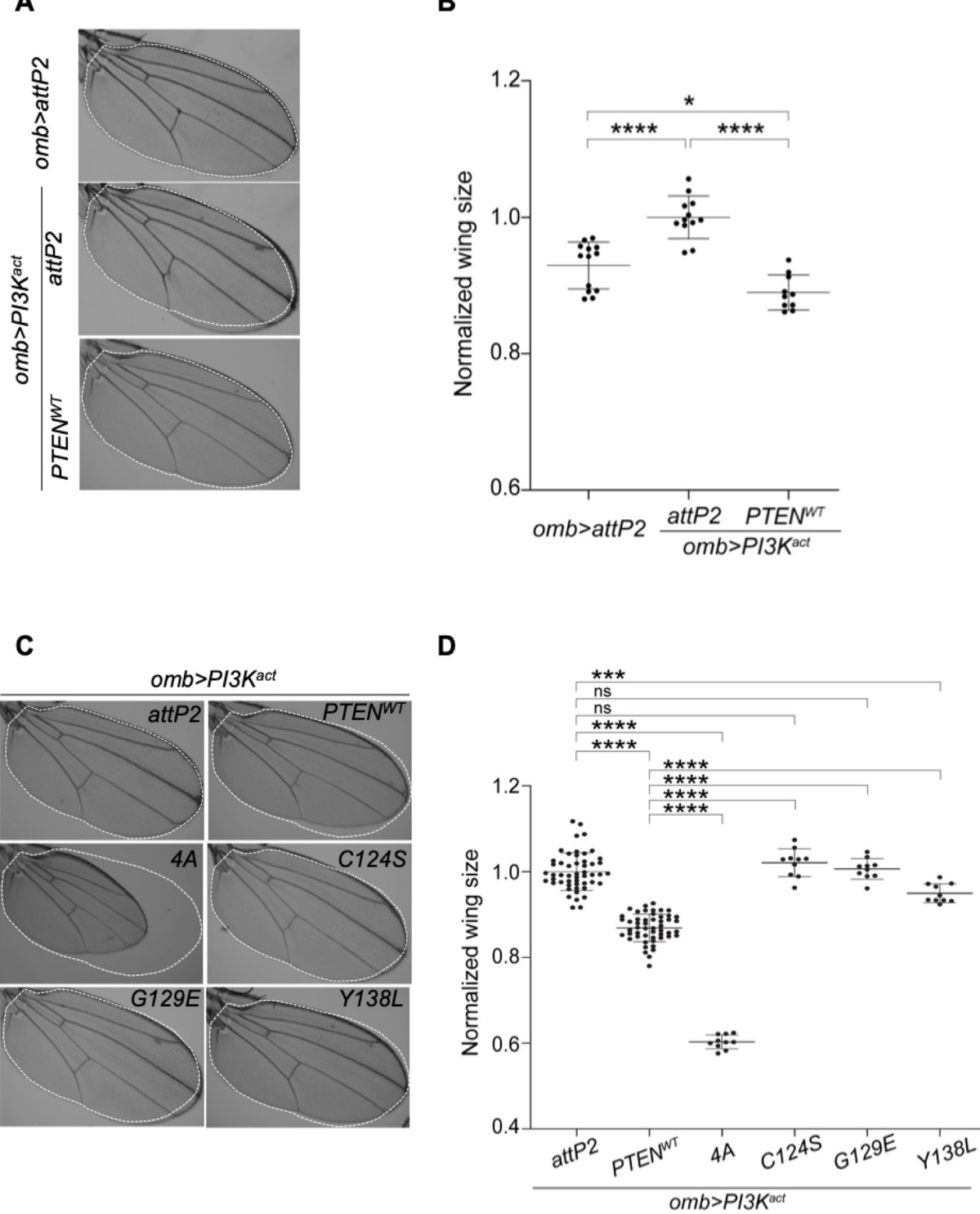

**Fig 2. Human PTEN suppresses PI3K-induced wing growth in *Drosophila*.** *UAS-PI3K^act* and *UAS-PTEN* variants were crossed to the wing pouch-specific GAL4 driver (*omb-GAL4*) to test wing growth. (**A**) Representative image of adult wing of *omb-GAL4* crossed to *attP2* parental strain. (**A, C**) Representative images of adult wings of flies expressing *UAS-PI3K^act* (*omb>PI3K^act*) plus a *UAS-PTEN* variant or *attP2*. (**B, D**) Graphs showing normalized wing size for each genotype shown. Expressing PTEN-WT suppressed PI3K^act induced wing overgrowth. Expression of 4A reduced wing size. C124S and G129E acted as loss of function variants, and Y138L functioned as a partial loss of function. Each datum point within scatter plots represents a single adult wing. Data are expressed as mean ± SD and analyzed using one-way ANOVA with Tukey HSD post-hoc; * p < 0.05, ** p < 0.01, *** p < 0.001, **** p < 0.0001. ns = not significant.

PI3K/Akt pathway activity impacting tissue growth in a phosphatase-dependent manner, but at a much greatly reduced level compared to Pten.

## Human PTEN suppresses *PI3K* induced proliferation in developing *Drosophila* wing discs

We further characterized the effects of *UAS-PI3K^act* expression by examining growth of the developing wing imaginal disc. We marked cells in which transgenes were expressed with *UAS-GFP*. This served as a proxy for cellular growth or proliferation, since changes in the size of the GFP marked region could reflect increases in either parameter. The *omb-GAL4* driver is expressed in a wide domain that encompasses the wing pouch as well as a portion of the presumptive hinge region (Fig 3A). We examined discs from female *omb>PI3K^act* larvae. Compared to *omb>attP2* controls, PI3K^act expression caused a significant increase in the size of the GFP expression domain (Fig 3B, 3C and 3E). This increased size was suppressed to control (*omb>GFP*) size by co-expression of PTEN-WT (Fig 3D and 3E). Biochemical variants C124S and G129E failed to suppress the PI3K^act-induced increase in the GFP domain, whereas Y138L was partially but not fully suppressive, as was expected since it has an intact lipid phosphatase activity. In contrast, 4A induced a significant reduction in GFP expressing domain beyond control size in the presence of PI3K^act expression. These data are consistent with the impact of these variants on adult wing size, shown in Fig 2D.

PI3K/AKT signalling had been shown to promote wing growth by increasing both cell size and proliferation [54, 70]. To determine the relative contribution of these in our model, we first examined cellular proliferation using immunoreactivity to phospho Histone 3 (PH3), a marker of cell mitosis. Expression of PI3K^act increased proliferation significantly (Fig 3C and 3F), and this was suppressed to control levels by co-expression of PTEN-WT (Fig 3D and 3F). A significant failure to suppress this increase in the proliferative marker was observed for the C124S variant only, while 4A resulted in a significant reduction in proliferation. However, the G129E and Y138L variants failed to exhibit any substantive difference from *attP2* controls and PTEN-WT. We reason that this is due to the high variability in the number of PH3+ cells observed within each genotype, leading to a substantial overlap of the data, even between *attP2* and PTEN-WT. This may be due to variability in the level of active mitosis captured at the time of fixation, in a tissue that proliferates over multiple days. We conclude that enhanced proliferation is a contributor to PI3K^act-induced increase in wing disc size, but that this assay is not, by itself, sufficiently robust to assess differences in variant function.

To determine whether increased size of the imaginal wing disc and adult wing results from increased cell size, we counted individual wing hairs within the same fixed area of adult wings. As each cell in the wing blade produces a single hair, the density of hairs in a fixed area can be used as a proxy to determine cell size. When PTEN-WT was co-expressed with PI3K^act, the number of hairs within a fixed area was not significantly different, even though the wing size was significantly smaller, which suggested no change in cell size, but a change in cell number (S2 Fig). Thus, in this context, expression of PI3K^act enhanced proliferation, and this effect was suppressed by co-expression of PTEN-WT. Three of the biochemical variants also showed no change in cell size in the *omb>PI3K^act* background (S2 Fig). In contrast, the 4A hypermorphic mutant significantly reduced cell size and increased hair count per unit area, showing that cell size can also be impacted by PTEN, but this is only clearly evident for variants with potently abnormal function.

## Human PTEN suppresses PI3K-induced *Akt* signaling

Next, we examined a readout of the PI3K signaling pathway following expression of wildtype PTEN or biochemical variants. PI3K catalyzes the conversion of $PIP_2$ to $PIP_3$ and leads to the

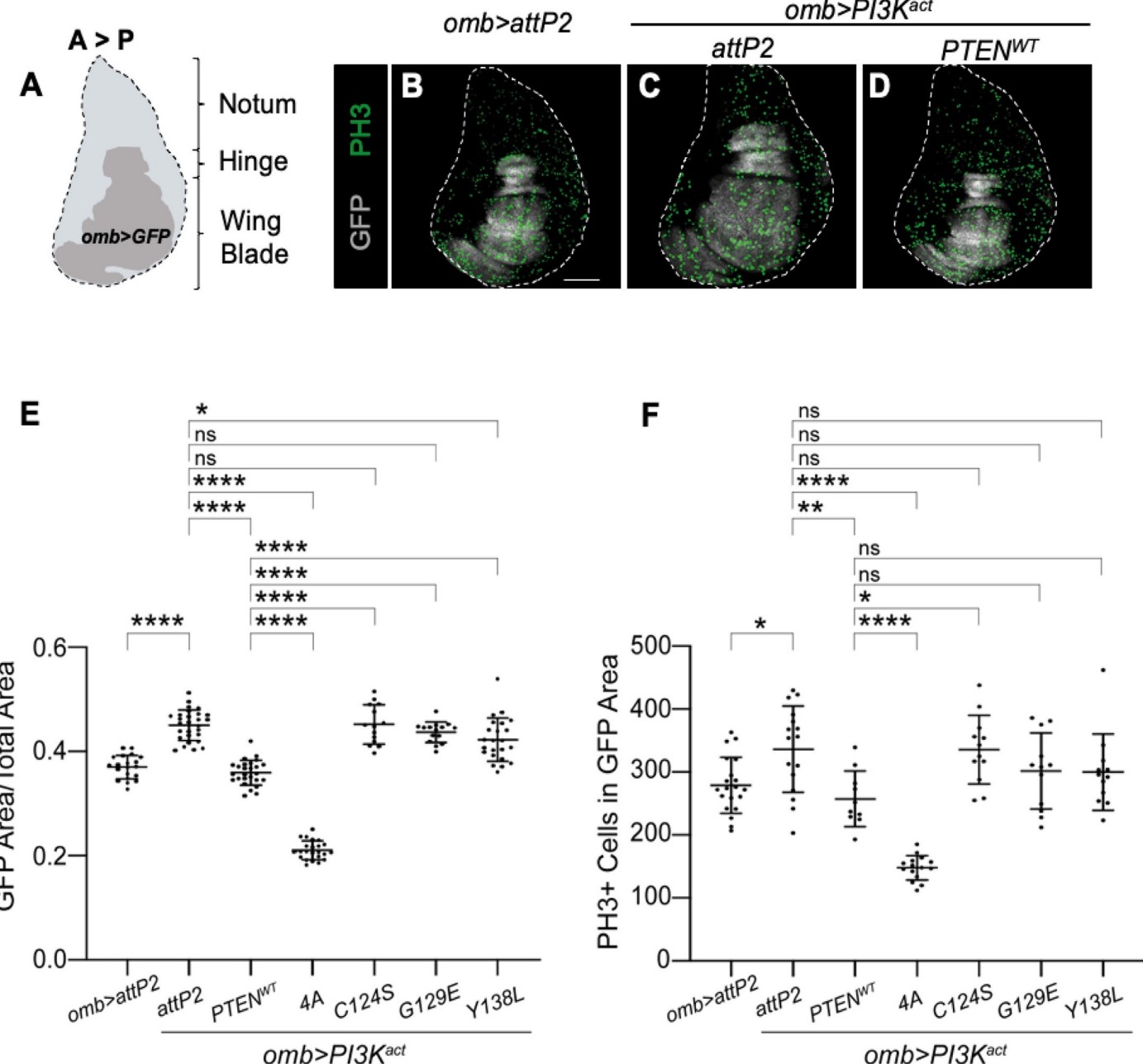

**Fig 3. Human PTEN suppress PI3K-induced proliferation in *Drosophila* wing imaginal discs.** Wing pouch specific driver *omb-GAL4* expressed *UAS-PI3K^act* and *UAS-PTEN* variants in the developing wing disc. Illustration of a *Drosophila* imaginal wing disc showing regions that develop into specific adult tissues after metamorphosis. The anterior to posterior axis (A>P) and the *omb-GAL4* expression domain (*omb>GFP*) are shown (**A**). Representative images of wing discs of 3rd instar female larva, with the wing pouch area marked by GFP (in grey) and PH3+ cells (in green) stained with anti-PH3 antibody for genotypes as shown (**B–D**). Graph showing ratio of GFP area over total area of the disc (**E**) and PH3 positive cells (**F**) in wing imaginal discs of each genotype. Expressing PTEN-WT reduced PI3K^act induced proliferation. Each datum point in scatter plots represents a single wing imaginal disc. Data are expressed as mean ± SD and analyzed using one-way ANOVA with Tukey HSD post-hoc; $p > 0.05$, * $p < 0.05$, ** $p < 0.01$, **** $p < 0.000.1$. ns = not significant. Scale bar is 100 μm.

phosphorylation of Akt by Pdk1 [37, 41–44]. We assessed pathway activity by examining immunoreactivity to phosphorylated Akt (pAkt S505) [71–73]. For these assays, we expressed *UAS-PTEN* in the posterior compartment of developing wing discs, using *engrailed-GAL4* (*en-GAL4*). This approach allowed comparisons of PI3K^act expressing posterior cells with

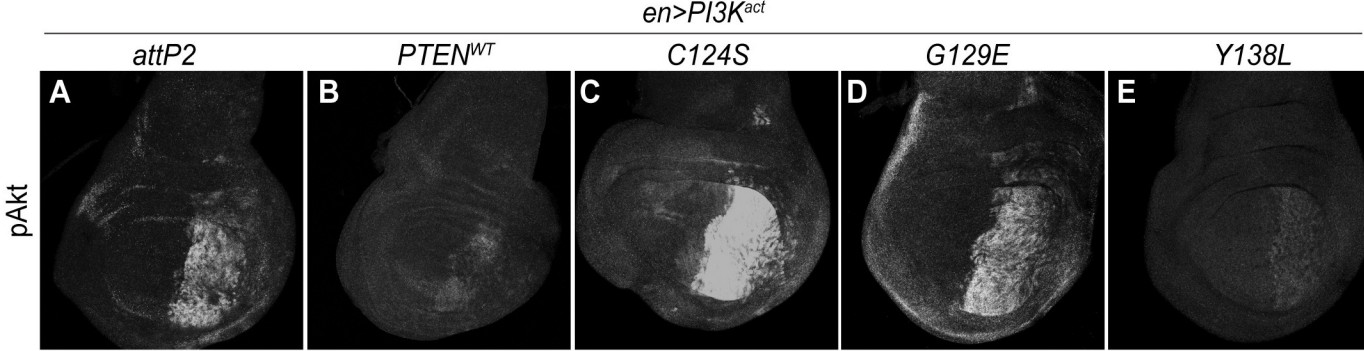

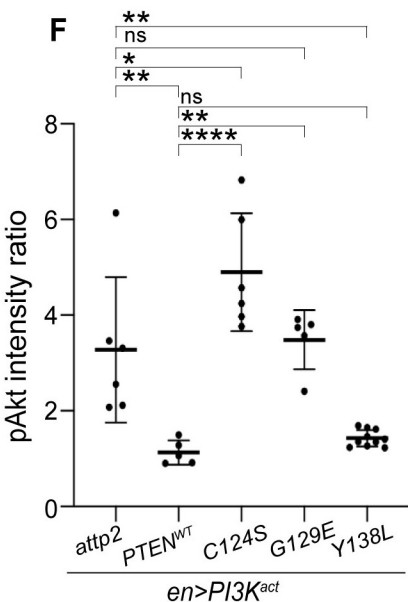

**Fig 4. Human PTEN suppresses PI3K-induced Akt signaling.** The posterior wing-restricted *en-GAL4* driver expressed *UAS-PI3K^act^* and *UAS-PTEN* and variants in the posterior half imaginal wing disc exclusively. Representative images of wing imaginal discs of 3^rd^ instar female larvae stained with anti-pAkt (**A-E**) for genotypes shown. (**F**) Graph showing pAkt posterior/anterior intensity ratio in the wing discs of each genotype. Expression of PTEN-WT and Y138L but not C124S and G129E restored pAkt levels. Each datum point within scatter plots represents a single wing imaginal disc. Data are expressed as mean ± SD and analyzed using one-way ANOVA with *post-hoc* Tukey HSD; p > 0.05, * p < 0.05, ** p < 0.01, **** p < 0.000.1. ns = not significant. Scale bar is 20 μm.

unaffected anterior regions, within single wing imaginal discs. We crossed *en>PI3K^act^* flies to *UAS-PTEN-WT* and three of the biochemical variants, *C124S, G129E* and *Y138L*. We were unable to assess the effects of *4A*, as its expression was lethal with *en-GAL4*.

S3 Fig shows the overexpression of PTEN (by immunoreactivity) in the posterior compartment of the developing 3^rd^ instar wing pouch *en>PI3K^act^*+PTEN-WT larvae. This figure also shows that *PI3K^act^* increases pAkt immunoreactivity and the levels of a PIP3 sensor, tGPH [74], both of which are suppressed by PTEN. When *en>PI3K^act^* flies were crossed to the control *attP2* strain, we observed significantly elevated anti-pAkt immunoreactivity in cells in the PI3K^act^ expressing posterior region (Fig 4A and 4F). We quantified the pAkt fluorescence intensity ratio of fixed areas within posterior versus anterior compartments of the late L3 wing imaginal disc. Consistent with the impact of PTEN on suppressing adult wing size, we found that *en>PI3K^act^*+PTEN-WT suppressed the enhancement of pAkt (Fig 4B and 4F and S3 Fig).

We also expressed the three biochemical variants and assessed the status of pAkt levels. C124S caused a significant elevation in pAkt intensity relative to *PI3K^act^+attP2* (Fig 4C and 4F). This may suggest interference with endogenous *Drosophila* Pten function, which may be consistent with observations of antimorphic activity for C124S using mouse genetic approaches [66]. G129E caused no change relative to *PI3K^act^+attP2*, indicative of a loss of function (Fig 4D and 4F). In contrast, Y138L resulted in a significant suppression of pAkt intensity (Fig 4E and 4F), mimicking the effects seen with PTEN-WT. This is consistent with observations that show that suppression of pAkt levels are unaffected by Y138L in other systems [58, 75].

## Functionalization of ~100 PTEN variants in suppression of *PI3K*-induced wing overgrowth

Having established that PTEN can suppress PI3K/AKT pathway in a wing growth assay, we proceeded to screen the relative function of PTEN variants from individuals with PHTS, somatic cancer, or ASD (S1 and S2 Tables), using the *omb>PI3K^act^* wing size assay, in addition to PTEN-WT, *attP2* (no *PTEN*) control, and the panel of biochemical variants (C124S, G129E, Y138L, 4A). We crossed homozygous male *UAS-PTEN* variant flies to *omb>PI3K^act^* females and assessed the wing size of adult female offspring (Fig 5). In brief, a minimum of 10 adult wings were quantified for each variant, and each wing was taken from a different adult female. Variants were tested in seven batches. To normalize for batch effects, we included PTEN-WT and *attP2* controls within each batch of crosses (S4 and S5 Figs). The results were plotted to show normalized wing size with respect to PTEN-WT wing size (at 1) and *attP2* wing size (at 0) (Fig 5).

In these tests, PTEN-WT and the biochemical controls reproduced their relative abilities to suppress *omb>PI3K^act^* wing size. Importantly, C124S and G129E demonstrated no PTEN activity, while Y138L showed around 27% of PTEN-WT activity. The hypermorphic 4A variant acted as a significant gain of function variant causing 374% of PTEN-WT function, and was significantly different from PTEN-WT when performing in-batch statistical comparisons.

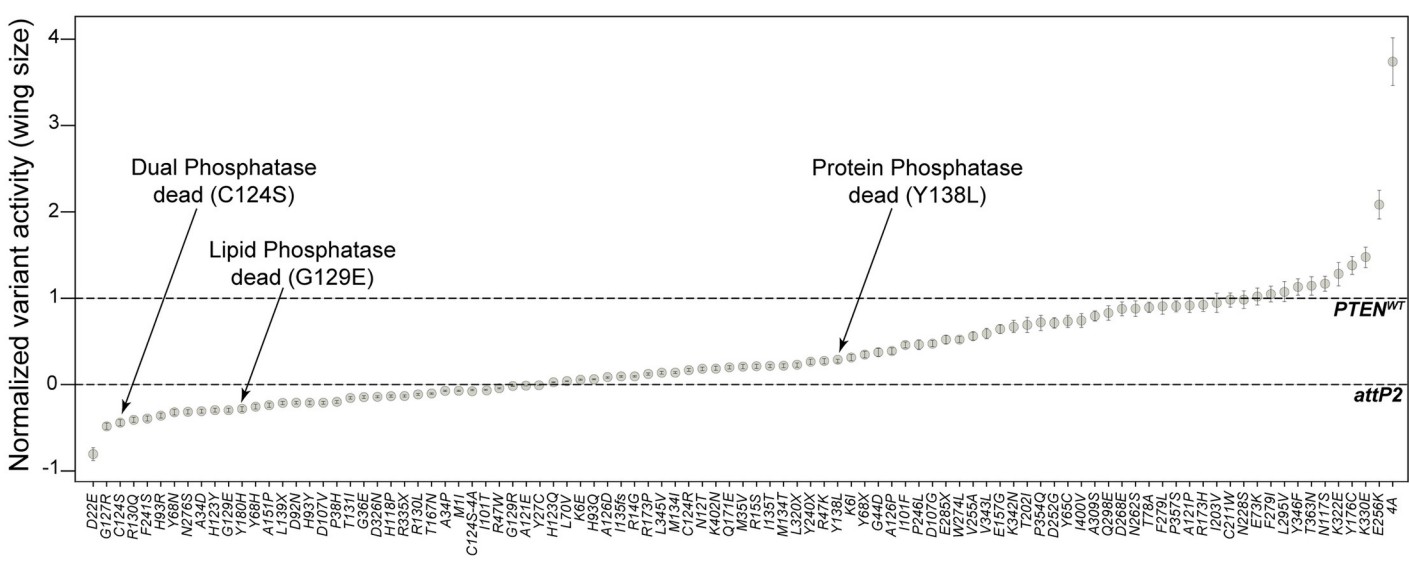

**Fig 5. Functionalization of human PTEN variants using the wing size assay.** 97 *UAS-PTEN* variant flies were crossed to *omb>PI3K^act^* females and progeny were assessed for wing size (see S1 Table). Graph showing normalized variant activity of PTEN variants where *attP2* (no *PTEN*) = 0 and PTEN-WT = 1. Each datum point indicates a normalized variant mean. Data are expressed as normalized mean. Error bars indicate propagated error (relative standard deviation).

All other variants showed a range of function across the spectrum of PTEN activity. To test the significant difference between variants, we employed one way ANOVA and a *post-hoc* Tukey test within each of the seven in-batch datasets (S4 and S5 Figs). Notably, these data showed that only G127R exhibited significantly less activity than *attP2*, suggestive of interference with endogenous PI3K/AKT signaling potentially through suppression of endogenous Pten activity. Additionally, both E256K and K330E variants were found to be gain of function in this assay. There are currently no studies to corroborate that these variants cause an increase in PTEN function in PI3K/AKT signaling, as seen in our assay.

### The fly wing assay accurately assigns PTEN variant function

We next assessed the relative accuracy of the wing size assay by comparing it to two other functional datasets from non-*Drosophila* models for the same set of PTEN variants. These included a HEK cell PTEN assay which is based on measuring relative pAKT/AKT immunofluorescence in the presence of endogenous PTEN [50], as well as a yeast assay that uses colony size as a readout of the lipid phosphatase activity of PTEN [58]. To determine how well the three datasets correlate, we performed hierarchical clustering of the normalized relative PTEN functional scores from the *Drosophila* wing, HEK and yeast assays (Fig 6A, raw and normalized data provided in S2 and S3 Tables). A heatmap comparing relative PTEN variant function for each assay demonstrated that all three assays generally produced similar functional readouts. However, we noted that the yeast assay assessed more variants as functionally normal than did the other two. Colour-coding of the hierarchical clustering branches was used to visualize groups of variants showing similar functional scores across the three assays (Fig 6B–6D). The branches denoted in yellow, dark orange and cornflower blue showed the greatest discrepancy as over-calling normal PTEN function for numerous variants in the yeast assay (Fig 6C and 6D), when compared to either wing or HEK assays.

To better compare these datasets, we calculated pair-wise Pearson's correlation coefficients (Fig 6B–6D). While the correlations were significant (p<0.0001) in each case, the *Drosophila* wing versus HEK assays had the highest (at r = 0.7318), compared to wing versus yeast (r = 0.4999) and HEK versus yeast (r = 0.6287). Notably, these correlations highlighted the variants overcalled as normal by the yeast assay, in the bottom right quadrant of Fig 6C and 6D. Finally, we compared the wing assay to the previous yeast SIM data for PTEN variant interaction with *vac14* mutants (r = 0.6718), and to the *Drosophila* time to eclosion assay (r = 0.6429) [50] (S6 Fig). It is notable that the wing and HEK assays exhibited the highest correlation between any two assays. A small number of variants were quite differentially assessed between the wing and eclosion assays, in particular Y180H, K322E in which the wing assay seems to be the outlier of most assays, and I135T, N276S, K330E in which the eclosion assay seems to be the outlier (S7 Fig). The specific reasons for these differences are unknown, likely reflecting some idiosyncrasy in each assay, perhaps impacting protein stability, localization, protein/substrate interaction, protein translation or post-translational modifications, or any number of other processes.

### Comparison of wing size scores to computational predictions

Computational prediction tools such as CADD and Polyphen-2 are used to efficiently predict the functional consequence of variants, *in lieu* of experimental assays [56, 57]. Here, we compared function calls for PTEN variants by the wing assay with CADD Phred and PolyPhen-2 scores. The core of the CADD algorithm uses a logistic regression model which outputs a continuous range of values [56]. CADD does not recommend cut offs to predict the function of a human gene variant. However, other methods such as GAVIN suggest that, for PTEN, CADD

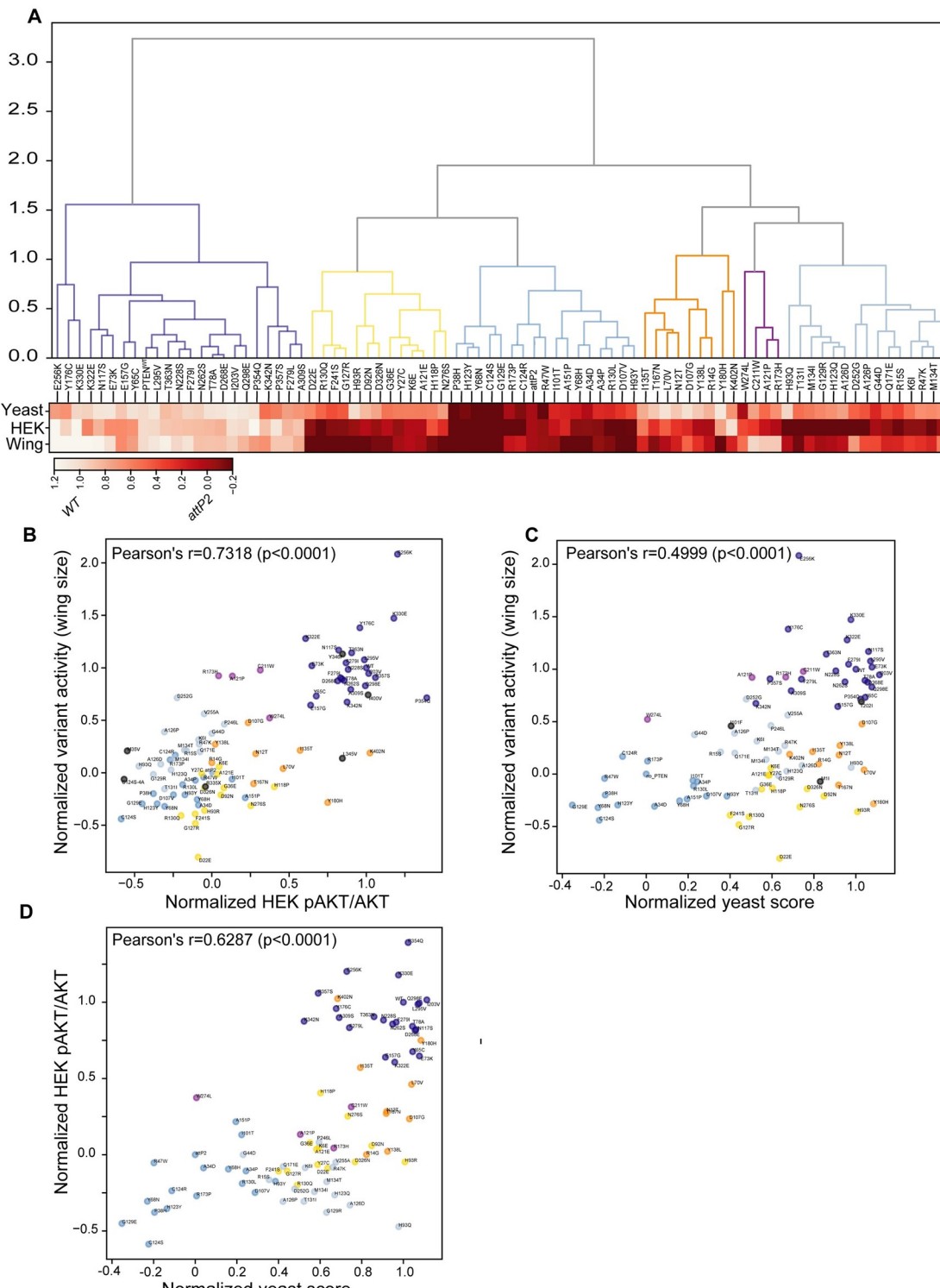

**Fig 6. PTEN variant function in the wing assay compared to HEK and yeast assays.** We tested the accuracy of the *Drosophila* wing size assay in predicting PTEN variant activity compared to available datasets assaying the ratio of pAKT/AKT immunoreactivity levels in HEK, or yeast growth dependent on $PIP_3$ to $PIP_2$ hydrolysis. (**A**) Hierarchical clustering to visualize functional similarities in normalized PTEN variant function from each assay, and a heatmap showing relative function. (**B-D**). Pairwise correlation scatter plots between the assays. Each datum point indicates an individual normalized variant mean (where *attP2* = 0 and PTEN-WT = 1) and each cluster is colour coded as in (**A**). A high correlation of the wing size and HEK assays support the accuracy of wing size assay

to accurately predict functions of PTEN variants *in vivo* across ~100 variants. (C, D) The over-calling of wildtype function for a set of variants (mostly those in the yellow, dark orange and cornflower blue clusters) is evident in the lower right quadrant of both plots.

Phred scores above 29.3 predict that the variant is damaging, and below 17.33 as benign. Using these criteria, only 13 of the 97 variants were assessed as damaging by this criteria, a significant under-representation of the variants found to be damaging according the wing and HEK assays [76] (Fig 7A). PolyPhen-2 uses a naïve Bayes model whose output is binary, and the scores are derived from the posterior probability. PolyPhen-2 classified many more variants as damaging than CADD (Fig 7B), however there were still 17 variants that disagreed with results from the fly assay; for example, ten loss of function (<50% of PTEN wildtype activity) variants were predicted to be benign (Fig 7B). Overall, we find that these computational methods disagree for many variants, and that the binary Polyphen-2 matches our experimental data better.

## Clinical PTEN variant interpretation using the *Drosophila* wing size assay

In order to assess the wing size assay for use in clinical interpretation, we identified known pathogenic and benign variant controls according to criteria established by the ClinGen PTEN Expert Panel, so that we could set readout thresholds for assigning variants as benign or pathogenic [12]. For pathogenic variants, we selected 10 variants that are PTEN Expert Panel curated as pathogenic in ClinVar. Identifying a sufficient number of classified benign variant controls was more challenging, as only one missense variant has been formally classified as likely benign in ClinVar [59]. Also, the ClinGen PTEN Expert Panel curated a list of benign and likely benign variants [36], but listed only one as a missense variant. Therefore, the

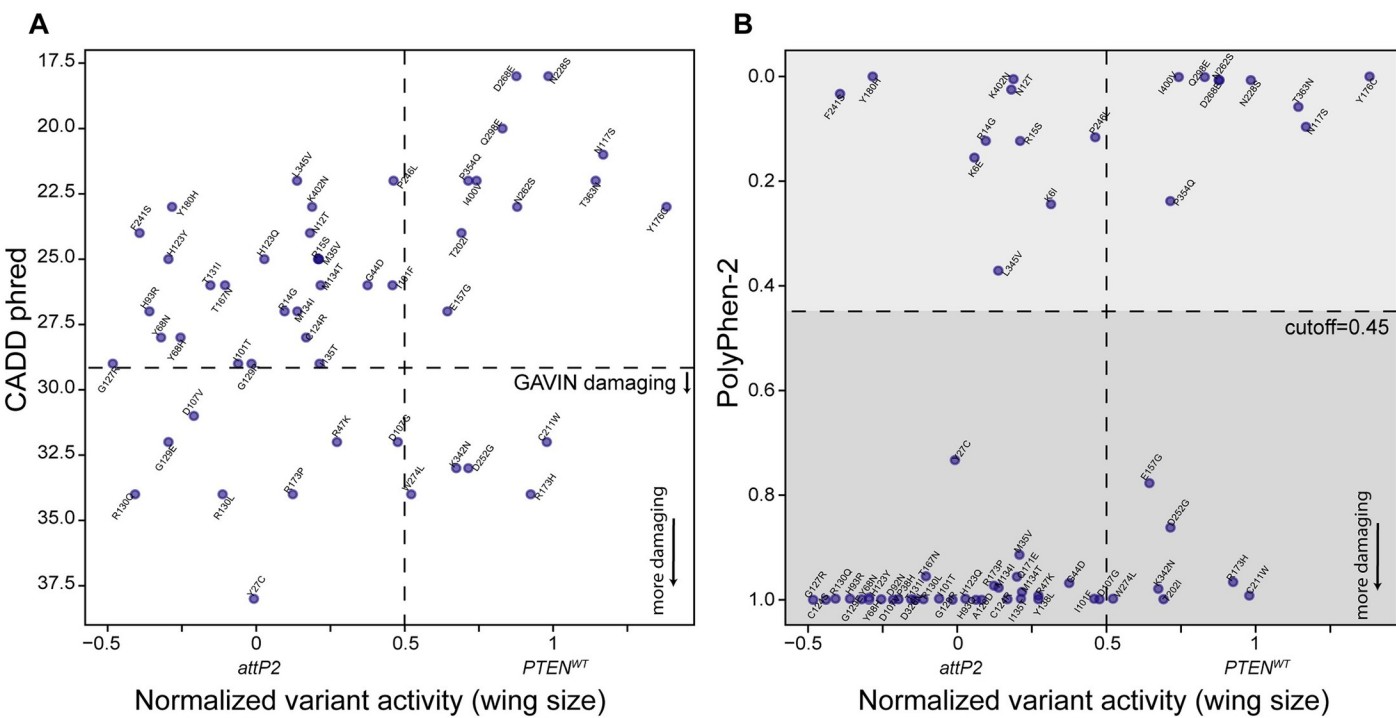

**Fig 7. Comparison of PTEN variant function in wings with computational predictions.** Graph showing normalized wing size data compared to CADD phred (**A**) and PolyPhen-2 (**B**). A score above 0.45 indicates a likely damaging variant by PolyPhen-2 and is shaded dark grey. Each datum point indicates an individual normalized variant mean (where *attP2* = 0 and PTEN-WT = 1).

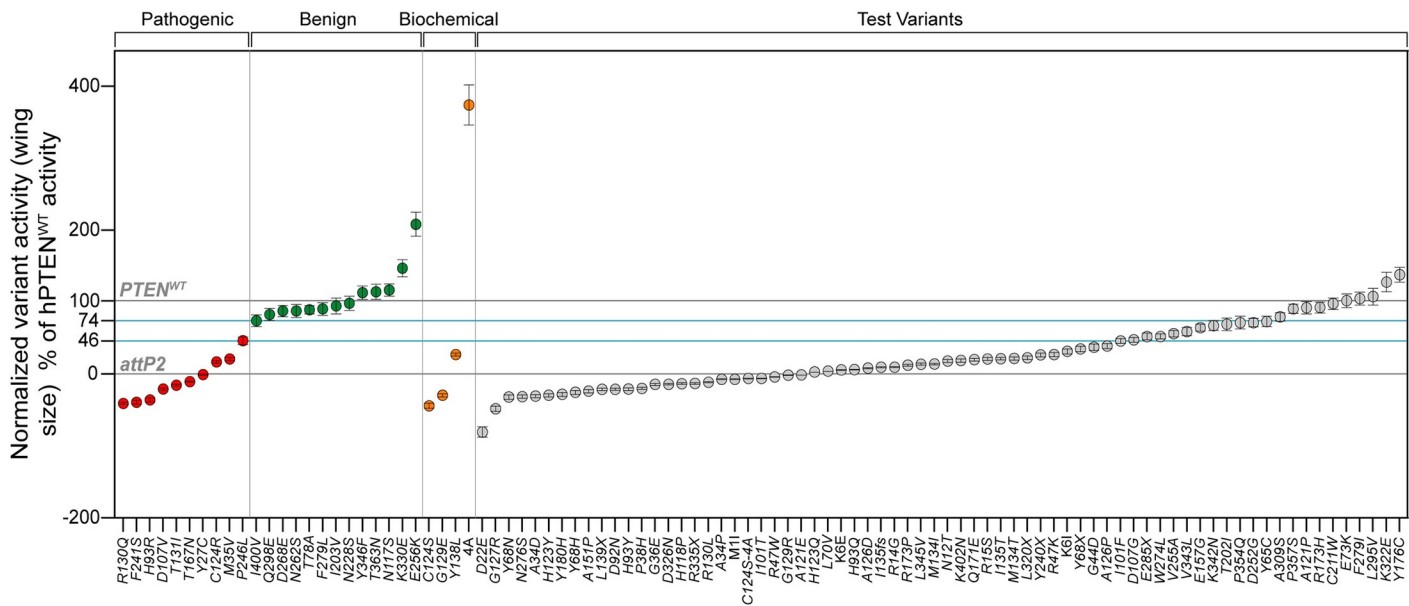

**Fig 8. Categorization of PTEN variants in wing size assay for clinical interpretation.** Assessment of the wing size assay for clinical interpretation was performed using the criteria established by ClinGen PTEN Expert Panel (see S2 Table). Normalized variant activity is shown as % of PTEN-WT activity (0 = *attp2* and 100 = PTEN-WT) calibrated using 10 pathogenic (all Expert panel reviewed in ClinVar), 13 benign (classified based on their normal function in three independent assays), and 4 biochemically-characterized variants. The assay successfully classified pathogenic and benign control variants into distinct functional groups concordant with their classified function. The additional 66 test variants were classified as either normal or benign (over 74% of PTEN-WT activity), abnormal or pathogenic (less than 46% of PTEN-WT activity) or intermediate or indeterminate (46 to 74% of PTEN-WT activity). The wing size assay successfully discriminated classified pathogenic and benign control variants into discrete functional readouts that permitted us to establish thresholds for clinical interpretation. Each datum point represents an individual PTEN variant. Data are expressed as mean. Error bars indicate propagated error (relative standard deviation).

majority were unsuitable for our study of coding variants. However, the ClinGen PTEN Expert Panel also approved the use of data from experimental assays, as long as two independent assays agreed to the normal function of the variant [36]. Based on these criteria, we were able to assign 13 variants as benign, based on their normal function in the three experimental assays [50, 58] (see S2 Table for details). Together, these pathogenic and benign variants exceed the 11 total known controls (comprising 6 pathogenic and 5 benign classified controls, or vice versa) required to satisfy criteria for clinical variant interpretation as either PS3_moderate (i.e. moderate functional evidence of being pathogenic), or BS3_moderate (i.e. moderate functional evidence for being benign).

Using the variant data from Fig 5, we replotted a graph that we calibrated with these pathogenic and benign variants (Fig 8). The wing size assay successfully discriminated these variants into discrete functional readouts, aligned to their classified functions. The pathogenic controls exhibited a loss of function of less than 46% of wildtype activity, whereas the benign controls exhibited 74% or more wildtype activity. Based on these criteria, we set readout thresholds for the wing assay as classifying variants as having functions that are normal (BS3_moderate), abnormal (PS3_moderate) or intermediate (indeterminate as either). Assessing all other test variants against these readout thresholds, we find that the majority are predicted as pathogenic. These results provide evidence that the wing size genetic interaction assay developed herein correctly distinguishes classified variants, thereby satisfying an important criterion for a well-established assay for use in clinical variant interpretation of test variants.

We wished to examine how well the current wing assay performed at functionalizing ClinVar pathogenic variants, compared to other assays; the *Drosophila* time to eclosion, HEK

pAKT/AKT ratio and yeast SIM assays [50], and the saturation mutagenesis yeast assay [58]. In S8 Fig we compare the relative function for each assay against three categories of pathogenic PTEN variants, expert panel curated, multiple submitters and single submitters. These data show that the wing assay performs very well across the 10 expert panel curated (also shown in Fig 8 for the wing assay alone) and 14 multiple submitters, except for one variant R173H. These data reiterate the tendency of the two yeast assays to over-assign pathogenic variants as being functionally normal. These data also establish that the wing assay correctly assigns abnormal function to 24 variants with better than single submitter evidence, and thus has a high positive predictive value for pathogenic variants.

The R173H variant is assigned a near normal function by all assays except the HEK cell assay. While these near normal function assignments stand in contrast to R173H's ClinVar multiple submitter pathogenicity annotation, this variant appears to be benign or functionally normal in many experimental assays assessing its function. In addition to R173H acting nearly functionally normally in two yeast assays (S6 and S7 Figs), it also acted normally in a premalignant mammary epithelial cells lacking *PTEN* (MCF10A cells) [77], and recapitulated a PTEN-WT gene expression signature for 978 landmark genes in HA1E cells, and failed to contribute to tumor formation in a mouse model [50, 77, 78]. Thus, we speculate that genetic context may play an important role in this variant's pathogenicity, which many assays fail to detect. It will be intriguing to follow further evidence for the pathogenic mechanism for this variant.

For the single submitter category, the wing assay closely paralleled findings from the HEK assay for 4 variants from strong loss of function to <50% of wildtype function. The T202I variant was reported pathogenic by a single submitter [79], but was assessed as near wildtype function in all assays in which it was examined, although unfortunately it was not assessed in the HEK assay. However, as a single submitter pathogenic annotation, there is currently insufficient evidence for a confident pathogenic annotation.

Interestingly, the variants E256K and K330E were defined as benign based on normal function in previous assays [50, 58], only K330E is reported in ClinVar as VUS. However, we found them to have significantly greater function than PTEN-WT in in-batch comparisons by one-way ANOVA and *post-hoc* Tukey comparisons. This emphasizes a potential advantage of the wing assay in its ability to determine whether a variant is gain of function. It will be interesting in the future to further verify any gain of function activity for these variants.

## Discussion

The goal of our study was to assess the use of *Drosophila* in well-established assays for scalable variant functionalization and clinical variant interpretation of human coding variants. We used PTEN as an example. We took a genetic interaction approach to test the primary defined gene-disease mechanism of PTEN variants, which exploits an established strength of *Drosophila's* tractable genetics to efficiently test a protein's function within specific pathways of interest. While formally possible that PTEN may function in a parallel growth pathway to counteract PI3K-induced growth, our combined data is most consistent with PTEN's interaction with the PI3K/AKT pathway in the *Drosophila* system. This includes our finding that *PTEN* rescues *Pten* hypomorphic overgrowth, and in the wing it suppresses PI3K induced increases in PIP3 and pAkt levels and also cellular proliferation and tissue growth. Together with evidence that the genetic interaction assay employed restores wing size to near wildtype size by suppressing a gain of function growth phenotype, and the extensive calibration with known variants, we reduce the possibility that our assay measures a neomorphic phenotype, any toxicity of the human protein, or any impact of PTEN on an independent pathway that also impacts wing

growth. We established an assay with a phenotype that is very easily scored in F1 progeny, after a single cross of a newly generated UAS-human variant. This makes the assay easily scalable to any number of variants, compared to gene replacement or rescue strategies requiring multiple crossings. The wing size phenotype also provides continuous quantitative data that directly and proportionately scores PTEN variant function in PI3K/AKT pathway-dependent tissue growth; this offers a linear readout of relative PTEN function across a wide range of variant activities, that we expect is less susceptible to threshold or saturation effects than indirect immunofluorescence or reporter readouts. This is also preferable to lethality assays for scoring ranges of variant activities which would tend to provide a more binary, or binned, output of loss of function (lethal) or normal function (viable). We validated the reliability of the assay by successful benchmarking against biochemically characterized variants, as well as expert panel curated pathogenic variants, and benign variants classified on approved grounds. This shows that the assay offers high positive predictive value for pathogenic variants, and high negative predictive value for benign variants; an important goal for a well-established assay. By setting readout thresholds following this calibration, we could determine the pathogenicity of 70 additional variants, according to ClinGen SVI guidelines. While these 70 variants have not been expert panel curated, we find excellent correlation in function for all variants between this *Drosophila* assay and a human cell line assay in our previous report [50], providing evidence of accuracy across this large number of variants.

We made two observations that may suggest potential interference of PTEN variants with Pten. Such 'dominant negative' effects are a potential artefact of overexpression of a loss of function variant, and thus may not represent evidence for antimorphic activity of the variant in a heterozygous state. However, there is considerable evidence for PTEN dimerism and for certain strong loss of function variants such as C124S acting in an antimorphic manner [66]. However, confirming an antimorphic effect of C124S or G127R on Pten would require detailed protein and functional interaction analysis between equimolar concentrations of the two, and is beyond the scope of this report. Instead, we believe it would be of more utility in the future to explore PTEN heteroallelic interactions in the *Drosophila* model.

There is no perfect assay for variant function, and often no perfect annotation for the function of many variants. First, no single assay can capture all aspects of the function of many proteins in disease, even in the case of a well understood protein like PTEN. The ClinGen PTEN Expert Panel considers evidence for <50% PTEN lipid phosphatase activity as evidence for pathogenicity (PS3) [36], and cancer resulting from reduced lipid phosphatase activity of PTEN is most often linked to reduced suppression of PI3K/AKT pathway activity [46, 80, 81]. However, lipid phosphatase-independent functions for PTEN have been described, including in the suppression of tumor progression related to regulation of p53 and cell motility, or gain of function truncating mutants in glioma [82–84]. Second, even when considering a single disease mechanism, a variant's impact in any patient arises from the integration of unknown synergistic or additive interactions between genetic, epigenetic and physiological states. This limits the predictive value of the variant classification in one individual for other unrelated individuals. Confidence is increased by observing similar effects of the same variant in unrelated individuals. Likewise, any assay designed to test multiple variants in any gene will have its own idiosyncrasies that can impact functionalization of certain variants, and all inevitably suffer from the loss of genetic, epigenetic and physiological context for each variant. Indeed, even models using cells or tissues derived from a single individual will suffer from culture artefacts, such as aberrant signal pathway activity, that could impact assessment of variant function.

Therefore, multiple different assays are recommended to build a consensus, and provide tolerance for each assay's failure to correctly assess a few specific variants [50]. In recognition

that assays would unlikely achieve 100% alignment with clinical classifications, the ClinGen SVI guidelines explicitly compensate for outlier datapoints in their odds of pathogenicity scores [12]. Over such large numbers of variants, there will inevitably be outliers due to idio-syncrasies of the assay or of the clinical classification. An example is R173H. This variant is annotated in ClinVar as pathogenic by multiple submitters, yet as outlined above has been shown to function normally or as a hypomorph in a range of different assays, including in the wing assay and in human tissues. We believe that instead of flagging any assays or clinical classifications as problematic based on such rare cases, such examples indicate that study of this variant's mechanism of action would likely yield interesting findings regarding PTEN function.

We propose that genetic models such as *Drosophila* offer tremendous potential for comple-menting other well-established assays for variant interpretation at scale. *Drosophila* provides a robust, *in vivo* genetic model that offers high reproducibility across technical and biological replicates. Cost-effective and efficient integrase-based transgenesis, the GAL4/UAS system, and the more recently developed MiMIC and CRIMIC gene replacement strategies offer unparalleled flexibility for human gene testing [23–26, 85]. Finally, a wide variety of genetic interaction tests can be easily developed, which with its long history of standard practice in *Drosophila* genetics is a well validated approach for explicitly testing specific disease mecha-nisms. Taking advantage of these many factors, there has been a surge of interest in assessing the function of human variants in *Drosophila* [19]. A major focus in such *Drosophila* studies has been to provide much-needed functional evidence for rare variants found in patients with rare diseases. Further analysis of gene and/or variant molecular activities can thereafter pro-vide insight relevant to diagnosis and therapy [19, 26, 86–88]. Such studies have been greatly augmented with support for consortia such as the Undiagnosed Disease Network (UDN) and the Rare Disease Models and Mechanisms Network (RDMM) [89–91], and accelerated by recent efforts to enhance the genetic toolbox for such studies [24–26, 88, 92]. The current study now adds large scale clinical interpretation of human variants to these efforts.

Which genes/proteins might be amenable to variant analysis in *Drosophila*? Clearly, the type of analysis reported herein requires a defined gene-disease mechanism and at least 11 known pathogenic and benign variants. This is not often the case; however, *Drosophila* has proven to be effective in discovering novel gene-disease mechanisms for genes with only few rare variants [19, 26, 86–88]. While many more genes will have to be tested in *Drosophila* to make firm conclusions, certain insights have emerged. A recent large-scale study found that human gene replacement of the *Drosophila* ortholog rescued lethality of an impressive 17 out of 34 gene tests (46%) [26]. This was similar to a study in yeast (1 billion years distant) showing that 47% of 414 essential genes could be replaced by their human ortholog [93]. This latter study examined enough genes to make a few fascinating observations. While sequence identity played some role in predicting replaceability, a short protein length and pathway-specificity proved to be the best predictor. Genes within the same pathway proved to be similarly replace-able, either most could be replaced in a pathway, or none could. Moreover, metabolic path-ways proved to be the most replaceable. There is no reason to believe that the ability to study human genes in *Drosophila* would not follow a similar pattern. It is notable that PTEN and Pten are only modestly conserved by global alignment (30.9% identity and 43.4% similarity by EMBOSS Needle [94]), but there is 61% identity by local alignment for the 109 amino acids comprising the phosphatase domain (data from DIOPT [95]). It is also notable that the main substrate for Pten in *Drosophila* is PIP3, a phospholipid that is identical in human, which we believe contributes to the success of this *Drosophila* assay. With the growing number of human genes being tested in *Drosophila*, as highlighted by a recent analysis of 79 human genes in one study [26], we will be in a better position in the future to assess which properties of a

human protein make it more tractable for functionalization in *Drosophila* assays. However, given the ~50% success rate in gene replacement in *Drosophila* seen in the only large scale analysis performed to date [26], and also the relatively low bioactivity of human PTEN compared to Drosophila Pten that we report herein, it is clear that we cannot assume that a human gene will act 'normally' in the *Drosophila* model, and great care should be taken to validate that the human gene is not acting in some neomorphic or toxic fashion outside of the mechanisms of interest. Regardless, our study demonstrates that a well-established *Drosophila* assay could provide a valuable addition to efforts to tackle the overwhelming gap that exists between variant identification and interpretation.

## Supporting information

**S1 Fig. Pten is more bioactive than PTEN in *Drosophila*, but both suppress PI3K-induced wing overgrowth. (A)** Representative adult wings from flies of genotype (from top to bottom), *omb>attP2*, *omb>PTEN* and *omb>dPten*. **(B)** Normalized adult wing size data in a non-PI3K activated background for flies expressing *omb>attP2*, *omb>PTEN*, *omb>dPten*, *omb>C124S*, *omb> G129E* and *omb> G127R*. Adult wing sizes normalized by dividing individual wing area data point by the average of *omb>attP2* wing area. **(C)** Representative adult wings from flies expressing (from top to bottom) *omb>PI3K^{act}+attP2*, *omb>PI3K^{act}+PTEN-WT*, *omb>PI3K^{act}+dPten*. **(D)** Normalized adult wing size data in a PI3K activated background for flies expressing *omb>attP2 (no PI3K)*, *omb>PI3K^{act}+attP2*, *omb>PI3K^{act}+PTEN*, *omb>PI3K^{act}+dPten*. Adult wing sizes were normalized by dividing individual wing area data points by the average of *omb>PI3K^{act}+attP2* area. Data are expressed as mean ± SD and analyzed using one-way ANOVA with *post-hoc* Tukey HSD; not significant (ns), $^{*}$ p < 0.05, $^{**}$ p < 0.01, $^{****}$ p < 0.0001.
(TIF)

**S2 Fig. Human PTEN suppresses PI3K-induced wing overgrowth by regulating cell proliferation. (A)** Representative image of an adult wing taken at 20X magnification (left panel) and zoomed in image to show the area in which wing hairs were counted, within the L3/L4 intervein region above the posterior cross vein. **(B)** Graph showing quantification of wings hairs for each genotype as shown. Each datum point in scatter plots represents a single wing. Data are expressed as mean ± SD and analyzed using one-way ANOVA with *post-hoc* Tukey HSD; $^{*}$ p < 0.05, $^{**}$ p < 0.01, $^{****}$ p < 0.0001. ns = not significant.
(TIF)

**S3 Fig. Expression of PTEN-WT in the posterior compartment of the wing disc supressed PI3K/Akt signalling.** Representative images of imaginal wing discs of $3^{rd}$ instar larvae stained with anti-GFP to visualize tGPH, a fluorescent sensor of PIP$_3$ levels [74] **(A,B)** anti-pAkt **(A', B')** and anti-PTEN **(A",B")** for *attp2* and PTEN-WT. Expression of PTEN-WT leads to suppression of both PI3K-induced PIP$_3$ and pAkt levels in the posterior compartment of the wing imaginal disc.
(TIF)

**S4 Fig. *Drosophila* wing size measurements, per experimental batch 1–3.** PTEN variants were tested across 7 batches total; *attp2* and PTEN-WT were repeated in each batch. **(A-C)** Adult wing size (in pixels) of PTEN variants assayed within each batch. Variants indicated as "GOF" had significantly smaller wings than PTEN-WT, "Like WT" were not significantly different from PTEN-WT, "Partial LOF" were significantly different from both PTEN-WT and *attP2*. "LOF" were significantly different from PTEN-WT and not significantly different from *attP2*. The wing size assay demonstrates the utility of *Drosophila* as a model system to test the

relative function of PTEN variants. Each datum point in the scatter plot represents a single adult wing. Data are expressed as mean ± SD. Significant differences as stated were obtained from analysis within each batch using one-way ANOVA with *post-hoc* Tukey HSD.
(TIF)

**S5 Fig. *Drosophila* wing size measurements per experimental batch 4–7.** PTEN variants were tested across 7 batches total; *attp2* and PTEN-WT were repeated in each batch. **(A-C)** Adult wing size (in pixels) of PTEN variants assayed within each batch. Variants indicated as "GOF" had significantly smaller wings than PTEN-WT, "Like WT" were not significantly different from PTEN-WT, "Partial LOF" were significantly different from both PTEN-WT and *attP2*. "LOF" were significantly different from PTEN-WT and not significantly different from *attP2*. The wing size assay demonstrates the utility of *Drosophila* as a model system to test the relative function of PTEN variants. Each datum point in the scatter plot represents a single adult wing. Data are expressed as mean ± SD. Significant differences as stated were obtained from analysis within each batch using one-way ANOVA with *post-hoc* Tukey HSD.
(TIF)

**S6 Fig. Comparison of PTEN variant function in adult wings with Δ*VAC14* and fly developmental delay scores.** Pairwise correlation scatter plots between the wing size assay and the yeast sentinel interaction mapping of PTEN variants using the Δ*VAC14* sentinel [50] (**A**), or the *Drosophila* developmental delay assay (time to eclosion) (**B**) [50]. Each datum point indicates an individual normalized variant (where *attP2* = 0 and PTEN-WT = 1) with clusters coloured according to clusters identified in Fig 6A.
(TIF)

**S7 Fig. PTEN variant function in the wing assay compared to developmental delay assay, HEK and yeast assays.** A heatmap comparing the relative function of PTEN variants in available assays; wing size assay in *Drosophila*, developmental delay assay (time to eclosion) in *Drosophila*, yeast growth dependent on $PIP_3$ to $PlP_2$ hydrolysis, Δ*VAC14* yeast sentinel and the ratio of pAKT/AKT immunoreactivity levels in HEK (where *no PTEN* = 0 and PTEN-WT = 1).
(TIF)

**S8 Fig. Comparing the performance of multiple assays for ClinVar pathogenic PTEN variants.** Graph showing normalized variant activity of ClinVar pathogenic PTEN variants where *attP2* (no *PTEN*) = 0 and PTEN-WT = 1, across 5 different assays. The pathogenic variants were grouped into three categories in ClinVar; expert panel curated, multiple submitters and single submitter. Each colour indicates a particular assay. Red circles indicate the wing size assay, blue squares indicate the eclosion assay, the black triangles indicate the HEK cell assay, the grey upside-down triangles indicate Δ*VAC14* yeast sentinel interaction of PTEN and the green diamonds indicate the yeast growth assay dependent on $PIP_3$ to $PlP_2$ hydrolysis. These data show that the wing assay performs very well across the 10 expert panel curated and 14 multiple submitters pathogenic variants. Each datum point indicates a normalized variant mean. Data are expressed as the normalized mean.
(TIF)

**S1 Table. Raw and normalized values for wing size assay.** 97 PTEN variants were assessed for wing size in 7 independent batches. Each batch comprised PTEN-WT, *attP2* and number of different variants. The raw and the normalized data for each PTEN variant assayed is provided.
(XLSX)

**S2 Table. Annotation of PTEN variants.** Each PTEN variant used in this study is listed with its ClinVar, COSMIC, gnomAD, normalized value for wing size assay, pAKT/AKT levels in HEK [50], $PIP_3$ to $PIP_2$ hydrolysis in yeast [58], VAC14 sentinel interaction in yeast and developmental delay assay in *Drosophila* [50], as well as CADD Phred and PolyPhen-2 scores. (XLSX)

**S3 Table. Raw values for each figure.** This spreadsheet provides the numerical value/quantification of images for all figures included in the manuscript. (XLSX)

## Acknowledgments

We thank Dr. William Gibson (BC Children's Hospital Research Institute, Canada), Dr. Kevin Eade (The Lowy Medical Research Institute, USA) and the Allan and Verheyen laboratories for useful comments and discussion regarding the manuscript. We are also grateful to Dr. Elizabeth Rideout (University of British Columbia, Canada) and Dr. Hugo Stocker (ETH Zurich, Switzerland) for critical genetic, molecular or immunological reagents. Flybase was used throughout these studies [96]. Stocks obtained from the Bloomington *Drosophila* Stock Center (NIH P40OD018537) were used in this study. We thank Dr. Kurt Haas (University of British Columbia, Canada), and Fabian Meili (Novartis Institutes for BioMedical Research, USA) for originally cloning the human PTEN variants [50], that we used to generate transgenic *UAS-PTEN* strains with the assistance of Rainbow Transgenics Inc (Camarillo, CA, USA).

## Author Contributions

**Conceptualization:** Payel Ganguly, Timothy P. O'Connor, Esther M. Verheyen, Douglas W. Allan.

**Data curation:** Payel Ganguly, Landiso Madonsela, Jesse T. Chao, Esther M. Verheyen, Douglas W. Allan.

**Formal analysis:** Payel Ganguly, Landiso Madonsela, Jesse T. Chao, Christopher J. R. Loewen, Esther M. Verheyen, Douglas W. Allan.

**Funding acquisition:** Christopher J. R. Loewen, Timothy P. O'Connor, Esther M. Verheyen, Douglas W. Allan.

**Investigation:** Payel Ganguly, Landiso Madonsela, Esther M. Verheyen, Douglas W. Allan.

**Methodology:** Payel Ganguly, Landiso Madonsela, Jesse T. Chao, Esther M. Verheyen, Douglas W. Allan.

**Project administration:** Esther M. Verheyen, Douglas W. Allan.

**Resources:** Christopher J. R. Loewen, Esther M. Verheyen, Douglas W. Allan.

**Software:** Payel Ganguly, Landiso Madonsela, Jesse T. Chao, Esther M. Verheyen, Douglas W. Allan.

**Supervision:** Timothy P. O'Connor, Esther M. Verheyen, Douglas W. Allan.

**Validation:** Payel Ganguly, Landiso Madonsela, Jesse T. Chao, Timothy P. O'Connor, Esther M. Verheyen, Douglas W. Allan.

**Visualization:** Payel Ganguly, Landiso Madonsela, Esther M. Verheyen, Douglas W. Allan.

**Writing – original draft:** Payel Ganguly, Timothy P. O'Connor, Esther M. Verheyen, Douglas W. Allan.

**Writing – review & editing:** Payel Ganguly, Landiso Madonsela, Jesse T. Chao, Christopher J. R. Loewen, Timothy P. O'Connor, Esther M. Verheyen, Douglas W. Allan.

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
