## [Decision Letter · Decision Letter 0]

4 May 2021

Dear Dr Allan,

Thank you very much for submitting your Research Article entitled 'A scalable Drosophila assay for clinical interpretation of human PTEN variants in suppression of PI3K/AKT induced cellular proliferation.' to PLOS Genetics.

The manuscript was fully evaluated at the editorial level and by independent peer reviewers. The reviewers appreciated the attention to an important topic but identified some concerns that we ask you address in a revised manuscript

We therefore ask you to modify the manuscript according to the review recommendations. Your revisions should address the specific points made by each reviewer.

[LINK]

Yours sincerely,

Norbert Perrimon

Associate Editor

PLOS Genetics

David Kwiatkowski

Section Editor: Cancer Genetics

PLOS Genetics

Reviewer's Responses to Questions

**Comments to the Authors:**

Reviewer #1: Next-generation sequencing is accelerating the discovery of genetic variants in clinical practice, yet the functional test and interpretation of these variants in a context that is clinically relevant remain a challenge.

They provide a proof of concept that Drosophila melanogaster may offer a robust platform for functional testing variants at scale and validated it for accurately assess the function of a large number of hPTEN variants.

The scalable in vivo assay used the Drosophila wing for screening ~ 100 human PTEN variants for their capacity to suppress a constitutively active form of PI3K.

This essay takes advantage of the conservation of PI3K, PTEN and AKT in flies, and the reproducible impact of this pathway on the growth of the development of Drosophila tissues.

The authors demonstrate that hPTEN functionally replaces Drosophila ortholog Pten and that similar to fly Pten overexpression, hPTEN suppresses normal wing growth as well as the wing overgrowth caused by gain of PI3K in various wing domains/compartments.

They used well biochemically characterized hPTEN mutants to validated the assays and then apply the design to classify 93 hPTEN variants. Moreover, they show that variant function calls are highly correlated with published data from a human HEK cell line. While this conclusion is fine, it raises the question of the true potential of Drosophila assays as an alternative.

Other comments are that the conclude that this assay can provide functional data for the interpretation of clinical variants for a large number of variants. However, the study provides a case for PTEN, but it lets open and there is not discussion about how generalizable and extensible is what they present to human proteins that are less either less conserved or have poor or completely unknown functions. That is the real challenge that the authors point out at the beginning of their introduction.

My second comment/criticism relates to the statement of “tests the disease-relevant physiological context and biochemical activity of PTEN”.

Despite this claim, what the authors do are tests of overexpression and hence supraphysiologic expression driven by the Gal4/UAS system and in the presence of two copies of endogenous PTEN, which is highly functionally conserved and which in my opinion combined with overexpression can be confounding factors for evaluation of certain type of mutations.

An assay/test that is missing and easy is to test the hPTEN variants to modify PI3KAct in the absence of endogenous Pten. This is relatively easy to do by starting the assays by constructing a stock with the gain of PI3K along with fly Pten RNAi or dCas9 / CRISPR and then express the hPTEN variants. I was surprised that this test has not been implemented,

Another missing test/assay that the authors could have explored is a knock-in in which the fly Pten is replaced by the hPTEN, so that researchers can assay function in a more physiological context without the overexpression resulting from the Gal4/UAS.

A knock in seems a strategy that may be more predictive and less prone to overinterpretation of, for example, gain of function. Clearly this is more time consuming but it could be discussed in the Discussion section as a future tool/approach to avoid supranormal level of human proteins which may in some cases be toxic to Drosophila —personal experience.

At the end of the discussion, the authors persuasively argue for the need to implement different assays to study variant function in different cell contexts, model organisms, cell cultures, and three-dimensional organoids. I fully agree and I think the study is convincing and well done. However, this message was not obvious and I found it confusing, but I might be wrong: my impression was that the goal has been to establish Drosophila as "the" system scalable to test batches of human variants together in the same setting but it seems that the goal is to use Drosophila assays as an additional toolkit. Perhaps clarifying these ideas of the discussion in the introduction would work better. I also missed a more critical evaluation of the advantages but also the limitations of the Drosophila paradigm based on the data obtained but also thinking on less conserved or conserved but less well functionally characterised human proteins that may be relevant for clinical interpretations and of interest for the non-specialists to evaluate the usefulness of Drosophila .

Reviewer #2: In the manuscript, Ganguly et al., establishes a Drosophila based functional assay to test the function of clinically identified variants in human PTEN (hPTEN). In a previous study (Post et al., 2020 Nature Comm), the same authors showed that one can classify hPTEN variants into functional categories (e.g. loss-of-function, partial loss-of-function, gain-of-function, benign) by over-expressing hPTEN in a wild-type fly and assessing its developmental timing and lethality. Because this assay was not scalable and somewhat indirect, the authors decided to come up with a better system that has more relevance to the patients’ condition, especially focusing on the function of hPTEN as a suppressor of PI3K/AKT activity. In this paper, the authors first showed that hPTEN can rescue the overgrowth phenotype of fly Pten mutants, demonstrating that these genes have conserved molecular functions. Next, they showed that expression of hPTEN can suppress the overgrowth phenotype caused by activated PI3K expression (PI3Kact) in the wing. After confirming that four PTEN variants with known biochemical consequences (C124S, G129E, Y138L, 4A) behave as expected in this wing-based assay (in both adult wing as well in the larval wing imaginal disc, tested parameters such as pAkt, cell proliferation, cell size etc.), the authors scaled up their assay to test 93 clinically relevant variants (generated through their previous study in Post et al., 2020). The authors concluded that their assay can pick up functional variants that were annotated as benign yeast assays, indicating that this could be a more sensitive assay. The authors also compared their results to in silico predictions and found that only a subset of functional variants identified in the fly system was predicted to be damaging. Finally, the authors came up with a diagnostic criteria of hPTEN variants using their functional data and the ClinGen PTEN 323 Expert Panel. By setting a phenotypic threshold of the wing phenotype to classify hPTEN variants into distinct severity (normal, abnormal, intermediate), the authors found that majority of the variants tested from the human cohorts with potential PTEN-related disorders were classified as pathogenic.

Overall, I feel the work to be of interest to the readership of PLoS Genetics, especially for clinical geneticists who are interested in VUS interpretation as well as Drosophila geneticists who study human diseases and genetic variants. The work is well designed and carried out with rigor. The manuscript is well-written and I feel there was a lot of work went into this study. Although the paper focuses on PTEN variants, a similar methodology can be used to study many variants in. diverse human disease relevant genes. I only have a handful of minor concerns before recommending this to be published.

Major Concerns

None.

Minor Concerns

1) Based on the Methods section, it seems that the transgenes used to test 97 variants (93 from patients and 4 biochemical) were reported in their previous study (Post et al., 2020 Nature Comm). However, I don’t see a direct comparison of the wing assay used in this paper with their previous functional assay (lethality/growth based on overexpression in otherwise wild-type animals). Were all the results consistent, or were there any outliers between the two fly assays? Since they do a comparison between their fly wing assay, yeast assay, human cell based assay and in silico predictions, I feel it is worth discussing what variants were missed in the previous growth based assay that is not picked up in the wing assay and vice versa.

2) I feel the rational of why the authors feel there is a need to develop a fly based assay are explained in diverse places within the paper, but I feel it would be nice to consolidate/re-emphasize it in one or two paragraphs in a logical fashion. I feel the section in the discussion that starts with Line 387 “No single assay can capture all aspects of the entire functionality of all PTEN variants, even in human cell or organoid models…” and Line 415 “First, each assay or model has relative strengths and weaknesses, and each assay will assess distinct…” has very important messages and explains that the assays that the author developed is complimentary to the other functional assays. By emphasizing these points more in their text, I feel that many more readers will recognize the significance of their research.

3) Line 352: change “that any benign variant” to “than any benign variant”

Reviewer #3: The manuscript by Ganguly et al describes the development of a Drosophila wing growth assay to assess the activities of different human PTEN variants. The authors nicely show that human PTEN can rescue loss-of-function for fly PTEN and that overexpressing human PTEN in the wing pouch of the developing imaginal disc can suppress the effect of overexpressing activated PI3K. They also assess the underlying changes in proliferation, cell growth and Akt signalling. Using this assay, nearly 100 different PTEN variants are assessed for activity and evidence is presented that this approach can score pathogenic versus benign mutations.

Overall, I think the authors have done a good job in characterising the assay and demonstrating that it has value in assessing the activity of novel variants. I am not a clinician, so I am not in a position to confirm how useful this will be, but my impression is that it will provide a robust assay that will complement other types of test in human and yeast cells.

I think there are a couple of additions that are required to complete the presented data, but I believe these will not be difficult to produce.

1. For Fig. 2, it would have been informative to see how the effect of overexpressed human PTEN in the wing compared to Drosophila PTEN, so that it could be determined whether the human protein activity is similar to the activity of fly PTEN in Drosophila. Has this been done?

2. On page 13, the authors propose that the C124S mutant may interfere with normal PTEN function to boost PI3K signalling. This is an interesting and important idea, and could be tested by overexpressing this mutant, the wild type protein and a protein with no apparent activity (G129E) in the absence of activated PI3K. It would also be useful to test the novel putative interfering mutants, G127R and D22E, to confirm that they behave in the same ways, since identification of such mutants has been flagged as one of the advantages of the fly system. Has this putative dominant negative activity been suggested in any other system?

Minor points

1. Page 9 – in the comment on Fig. 2A, B (line 160), it seems that overexpressed hPTEN actually reduces growth below wild type levels, rather than just fully suppresses the activated PI3K phenotype. This should be mentioned because it suggests that the 100% activity value involves a subtle reduction in wing growth when combined with activated PI3K, a weaker version of the 4A mutant’s effect.

2. Typos: line 251 – endogenous

Line 331 – our study

Reviewer #4: Ganguly and coworkers demonstrate the utility of the Drosophila system for functionally characterizing and categorizing genetic variants of the human PTEN gene, many of which are of unknown functional significance. This study thus adds the fly system to existing such assays that use human or yeast cells. The authors test the ability of wild type human PTEN and three biochemically characterized variants to suppress a number of phenotypes caused by activated PI3K, finding that suppression of a wing overgrowth phenotype provides a reliable readout of PTEN function. Further characterization of a series of validated pathogenic or predicted benign PTEN variants allows them to define thresholds of activity in this assay, demonstrated the potential utility of this system as an in vivo model to predict functionality of variants of unknown significance.

Generally, the experiments are well controlled and described, and the authors demonstrate how this system addresses the criteria as outlined by the working groups and panels that have highlighted the need for functional assays. However, the reliance on GAL4-mediated over-expression, both of PTEN and of PI3K, may undermine the in vivo relevance of these results. Additionally, the authors fail to demonstrate predictive power of their assay on an untrained data set, which further limits the applicability of their findings.

Specific comments

1. The assays described here measure the relative ability of PTEN variants to suppress the effects of overexpressed, constitutively activated PI3K, which leads to an enlarged wing when driven by the omb-GAL4 driver. This would seem to be an appropriate context for PTEN to function in suppressing oncogene-driven tumorigenesis, but likely does not reflect its action in cells with normal growth factor signaling. A focus on the ability of human PTEN variants to rescue Drosophila PTEN mutant phenotypes, such as pupal overgrowth, would be a more physiologically relevant test of variant function.

2. The GAL4-UAS-mediated overexpression of the normal and variant hPTEN alleles is also potentially problematic. Protein levels are not examined here, and this system typically drives expression at multiples of the endogenous gene. A large number of the tested variants behave as dominant-negatives in the wing assay, likely impacted by this high level of expression. The variants would need to be expressed at endogenous levels, in a heterozygous or homozygous dPTEN mutant background, to have informative predictive value. Minimally, testing and comparing effects at multiple expression levels is needed to convincingly demonstrate relevant function.

3. The authors show that selected groups of 14 pathogenic and 11 nominally functional variants segregate into discrete groups in the wing overgrowth assay, and they use these groups to set scoring thresholds to sort a collection of PTEN variants. In scoring these remaining test variants, however, they fail to assess whether these scores actually provide useful information – whether known pathogenic variants fall below the cutoff and benign variants above it. A cursory examination of several alleles suggests that this is not the case. For example, the R173H variant is categorized as pathogenic by ClinVar based on multiple lines of evidence, but it falls well above the benign cutoff as shown in Figure 8. Another pathogenic variant, T202I, lies just at the benign threshold. These examples would seem to call into question the confidence one could place in results based on this assay, and they highlight the need for further improvement of this model.

**Have all data underlying the figures and results presented in the manuscript been provided?**

Reviewer #1: Yes

Reviewer #2: Yes

Reviewer #3: Yes

Reviewer #4: None

PLOS authors have the option to publish the peer review history of their article (what does this mean?). If published, this will include your full peer review and any attached files.

Reviewer #1: No

Reviewer #2: **Yes: **Shinya Yamamoto

Reviewer #3: No

Reviewer #4: No

---

## [Decision Letter · Decision Letter 1]

10 Aug 2021

Dear Watt Allan,

We are pleased to inform you that your manuscript entitled "A scalable Drosophila assay for clinical interpretation of human PTEN variants in suppression of PI3K/AKT induced cellular proliferation." has been editorially accepted for publication in PLOS Genetics. Congratulations!

Please note the few minor revisions from one of the reviewer which we encourage you to address as needed.

Yours sincerely,

Norbert Perrimon

Associate Editor

PLOS Genetics

David Kwiatkowski

Section Editor: Cancer Genetics

PLOS Genetics

Comments from the reviewers (if applicable):

Reviewer's Responses to Questions

**Comments to the Authors:**

Reviewer #1: In my opinion the manuscript has improved considerably and the discussion has clarified and emphasised better the goals and the results. I think this work will be of great interest for the cancer research community

One note, since hPTEN has been changed throughout the text to PTEN, it is unnecessary in the abstract ( line 28) to include (PTEN) follow PTEN

Reviewer #2: I feel the authors have improved their text to clarify their points and added an important panel (data using fly Pten) in their revised document. I feel the paper is of relevance to the readership of PLoS Genetics.

Reviewer #3: The authors have addressed the comments in my original review. They have produced a quite substantive rewrite in several parts of the manuscript, which I think does help to better place the study in context and explain the motivation behind it.

Reviewer #4: The revised manuscript has been extensively rewritten, with an increased emphasis on how the results fit in the context of a defined disease mechanism, namely the interaction of PTEN with the PI3K signaling pathway. The utility of this system is also more appropriately described, as a complement to other assays. And some new data comparing the effects of human and fly Pten in this system are included.

Despite these improvements, in describing the effects of overexpressed Pten variants in the background of PI3K-induced overgrowth, the authors ascribe a specificity to these interactions that is not warranted. For example:

“A genetic suppression approach is preferred in genetic interaction tests because typically only genes acting within the upregulated pathway are expected to selectively suppress the resulting phenotype.”

“… the wing assay alone amongst these assays explicitly tests PTEN suppression of the PI3K pathway in tissue growth”

“Moreover, as the genetic interaction restores wing size to near wildtype size by suppressing a gain of function growth phenotype, we reduce the possibility that we are assaying a neomorphic phenotype or toxicity of the human protein, and also selectively test PTEN function to suppress the excessive signaling activity that exceeds the capacity of endogenous Pten.”

I don’t believe the authors demonstrate a selective suppression of the PI3K overgrowth phenotype. As shown in supplement, Pten overexpression also reduces the size of control wings; perhaps a comparison of the effect of Pten on PI3K-overexpressing vs control wing size would show selectivity, but this is not included. The acceptance of this assay would be strengthened by demonstrating such selectivity.

As a thought experiment, one can imagine that Pten overexpression would likely also reduce the size of wings enlarged by expression of Myc or other oncogenes; or that loss of these factors would reduce the size of PI3K-overexpressing wings. More to the point, a “neomorphic or toxic human protein” would be expected to suppress PI3K-driven overgrowth just as well as a specific inhibitor of the pathway.

Two growth regulators do not have to act through the same pathway to have counteracting effects on growth. The classic genetic interaction tests that have been useful in identifying pathway components typically involve heterozygous loss of function alleles that have no visible phenotype on their own. This is fundamentally different than the co-overexpression experiments described here. I think the authors need to be more circumspect in describing the potential specificity of their phenotypes.

Finally, the authors conclude that “PTEN can suppress endogenous PI3K/Akt pathway activity impacting tissue growth in a phosphatase-dependent manner, but at a much greatly reduced level compared to Pten.”

It should be made clear that this is true in the context of the Drosophila wing. Human PTEN may be as or more capable than fly Pten of suppressing the PI3K pathway in human cells.

**Have all data underlying the figures and results presented in the manuscript been provided?**

Reviewer #1: Yes

Reviewer #2: Yes

Reviewer #3: Yes

Reviewer #4: None

PLOS authors have the option to publish the peer review history of their article (what does this mean?). If published, this will include your full peer review and any attached files.

Reviewer #1: No

Reviewer #2: **Yes: **Shinya Yamamoto

Reviewer #3: No

Reviewer #4: No

**Data Deposition**

http://datadryad.org/submit?journalID=pgenetics&manu=PGENETICS-D-21-00433R1

**Press Queries**

---

## [Editor Report · Acceptance letter]

1 Sep 2021

PGENETICS-D-21-00433R1 

A scalable *Drosophila* assay for clinical interpretation of human PTEN variants in suppression of PI3K/AKT induced cellular proliferation. 

Dear Dr Allan, 

We are pleased to inform you that your manuscript entitled "A scalable *Drosophila* assay for clinical interpretation of human PTEN variants in suppression of PI3K/AKT induced cellular proliferation." has been formally accepted for publication in PLOS Genetics! Your manuscript is now with our production department and you will be notified of the publication date in due course.

With kind regards,

Agnes Pap

PLOS Genetics

On behalf of:
